# ⬡ From an LLM Swarm to a PDDL-empowered Hive: Planning Self-executed Instructions in a Multi-modal Jungle

**Kaustubh Vyas**[†]**, Damien Graux**[†]**, Yijun Yang**[†]**, Sébastien Montella**[†]**, Chenxin Diao**[†]
**Wendi Zhou**[¶ ‡]**, Pavlos Vougiouklis**[†]**, Ruofei Lai**[†]**, Yang Ren**[†]**, Keshuang Li**[†]**, Jeff Z. Pan**[† ¶]
[†] Huawei Technologies Ltd., UK    [¶] University of Edinburgh, UK
{firstname.lastname}@huawei.com  {wendi.zhou,j.z.pan}@ed.ac.uk

## Abstract

In response to the call for agent-based solutions that leverage the ever-increasing capabilities of the deep models' ecosystem, we introduce Hive – a comprehensive solution for knowledge-aware planning of a set of atomic actions to address input queries and subsequently selecting appropriate models accordingly. Hive operates over sets of models and, upon receiving natural language instructions (i.e. *user queries*), schedules and executes explainable plans of atomic actions. These actions can involve one or more of the available models to achieve the overall task, while respecting end-users specific constraints. Notably, Hive handles tasks that involve multi-modal inputs and outputs, enabling it to handle complex, real-world queries. Our system is capable of planning complex chains of actions while guaranteeing explainability, using an LLM-based formal logic backbone empowered by PDDL operations. We introduce the MuSE benchmark in order to offer a comprehensive evaluation of the multi-modal capabilities of agent systems. Our findings show that our framework redefines the state-of-the-art for task selection, outperforming other competing systems that plan operations across multiple models while offering transparency guarantees while fully adhering to user constraints.

## 1 Introduction

Within the past few years, the number of available models—either through commercial paywalls or open-sourced—has exploded both in terms of intrinsic performances and in terms of tasks handled by them, ranging from text generation (Achiam et al., 2023; Anthropic, 2023; Team et al., 2023; He et al., 2025) to more specific actions such as code generation (Becker et al., 2023; Dong et al., 2024) or image generation (Wang et al., 2023b; Zhu et al., 2023). This rapid growth has unlocked unprecedented potential for real-world applications, inspiring practitioners, especially in industry, to envision new use cases that leverage these powerful models (Liu et al., 2023c; Shen et al., 2024a; Lu et al., 2024; Xing et al., 2024). However, if creativity and possibility have been *unleashed* by such a surge, implementing pipelines that involve multiple models remains a complex and largely manual (and often cumbersome) process, particularly when addressing tasks beyond the original design of these models. This often leads developers to create ad hoc modules to manage these complexities. In addition, a significant number of models available in the wild are either advanced proof-of-concept or very specialised ones, see *e.g.* the hundreds of thousands of models available on the HuggingFace platform[1]. As a consequence of this abundance, navigating through this jungle to **select** the appropriate models for a set of tasks has become very challenging. This complexity arises both in terms of performance and compatibility. Connecting models' input and output formats is complex, as the generated results are often difficult to control (Scholak et al., 2021; Qin et al., 2022). Moreover, **planning** and chaining tasks for real-world use-cases present a significant challenge too.

In this study, we present a comprehensive solution to tackle the aforementioned two challenges, *i.e.* **(I)** selecting appropriate models and then **(II)** planning a set of atomic actions to achieve the objectives in the end-users' instructions (i.e., *user queries*). Our knowledge-aware planning system, **Hive**, takes natural language instructions (potentially involving multi-modal inputs and outputs)

---

[‡]Work done while at Huawei Technologies Ltd.
[1]1 016 247 models on `https://huggingface.co/models` as of October 1[st], 2024.

and can effectively schedule, execute and explain plans composed of atomic actions. These plans may involve one or more models, carefully orchestrated to accomplish the overall task while adhering to users-specific constraints, such as model size or licensing, to name a few.

One of our key contributions addresses the first challenge: the lack of *machine-understandable* interface that consolidates comprehensive information about available models. To bridge this gap, we propose a Capability Knowledge Graph (C-KG), a knowledge graph (Pan et al., 2017) with all dimensions needed to for automated planning and execution. For each model, *inter alia*, C-KG captures critical details such as supported tasks, performance metrics from state-of-the-art benchmarks, and minimal code snippets for inference. Additionally, to enable the planning of complex action sequences with guaranteed explainability, we developed a novel planning approach which, instead of relying solely on LLM reasoning capabilities (Pan et al., 2023), also employs formal logic to reach its conclusions. To achieve this, we took advantage of PDDL—a formal language widely used in robotics for defining planning problems (Aeronautiques et al., 1998), mapping the end-users' instructions with atomic actions, thereby enabling the conversion of natural language instructions into a PDDL problem space. This approach allows us to formally plan before executing the tasks using code snippets from the C-KG. As a result, it has enabled us to generate detailed reports that provide end-users with fine-grained and reliable explanations.

In the absence of standard publicly available benchmarks for solving real-world tasks, we introduce MuSE and share it as a Github repository[2], a new evaluation benchmark of complex queries involving multi-modal inputs and outputs, to assess our proposed framework. Using MuSE, we reviewed the closest existing solutions, namely HuggingGPT (Shen et al., 2024a) and ControlLLM (Liu et al., 2023c), which only tackle sub-problems of our broader objectives. The results indicate that HIVE not only surpasses these competitors but also consistently outperforms them across all benchmark dimensions. HIVE demonstrates a 30% higher accuracy in task selection and respects user constraints in 100% of cases, while being more reliable.

## 2 PRELIMINARIES

Planning Domain Definition Language (PDDL) (Aeronautiques et al., 1998) is a standardised language extensively used in the field of artificial intelligence (AI) planning to represent planning domains and problems. PDDL provides a formal syntax and semantics for defining the components of a planning task, including actions, predicates, objects, and their relationships. It enables the clear specification of the initial state, goal conditions, and permissible actions within a domain, facilitating the development and comparison of planning algorithms. In our work, PDDL plays a critical role in task decomposition and planning. By defining tasks as actions within PDDL domains, we leverage established planning techniques to generate coherent and feasible plans. The use of PDDL allows us to formally model complex tasks, ensuring that the system can reason about the preconditions and effects of actions within a well-defined framework.

Let $\mathbb{D} = \left\{ d_1, d_2, \ldots, d_{|D|} \right\}$ be a set of PDDL domains. Each PDDL domain $d_j \in \mathbb{D}$ is associated with a set of PDDL actions s.t. $\mathbf{a}^{d_j} = \left\{ a_1^{d_j}, a_2^{d_j}, \ldots, a_{A^j}^{d_j} \right\}$, where $j \in [1, |D|]$ and $A^j \in \mathbb{N}$ the number of actions included within the PDDL domain $d_j$. Furthermore, let $\mathbb{T}$ be a set of different tasks, consisting of all PDDL actions across the available PDDL domains $\in \mathbb{D}$, as follows: $\mathbb{T} = \bigcup_{j=1}^{|D|} \mathbf{a}^{d_j}$.

Finally, we define $\mathbb{M} = \left\{ m_1, m_2, \ldots, m_{|M|} \right\}$ as the set of all models available for completing a set of different tasks $\mathbb{T}$ or combinations thereof.

## 3 HIVE — GENERAL ARCHITECTURE

### 3.1 CAPABILITY KNOWLEDGE GRAPH

We extract *model cards*[3] directly from HuggingFace and incorporate an OpenIE extraction route for converting the textual descriptions from each model card into a structured representation. We align

---

[2] `https://github.com/dgraux/Hive-ICLR-2025`
[3] `https://huggingface.co/docs/hub/en/model-cards`

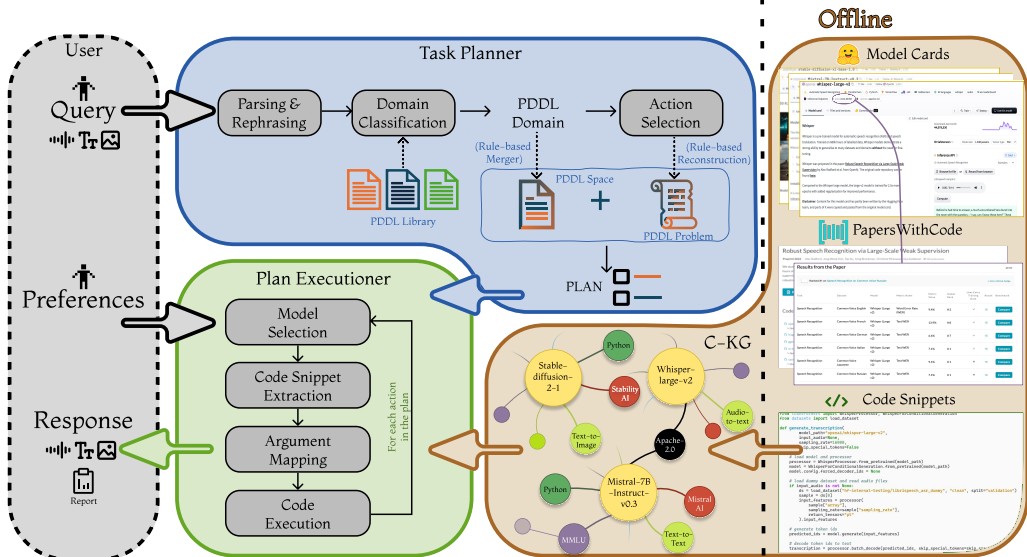

Figure 1: HIVE modular architecture.

the models with Papers With Code[4], which enables us to collect information about how a particular model performs across different benchmarks. Our goal is to build a Capability Knowledge Graph, denoted as C-KG, which is a graph $G(\mathbb{M}, \mathbb{T}, E)$, such that each model $m_j \in \mathbb{M}$ is associated with one or more tasks from $\mathbb{T}$ along with its relevant performance, and different types of edges $\in E$ with various properties (e.g., number of parameters or supported languages). We search each model card collected by HuggingFace for potential arXiv[5] paper ID. We make use of the arXiv paper ID in order to bridge models from HuggingFace with their corresponding performances in different tasks and benchmarks. When we identify an arXiv paper ID within the knowledge base of Papers With Code, we extract all model versions[6] and their performances across the different benchmarks.

Each different model version is represented as a separate vertex within the C-KG. A model vertex $m_j \in$ C-KG is aligned with its model card *if* the model's ID can be matched (e.g., `flan-t5-base`[7]) in both the original model card and the ID of at least one of the retrieved records from Paper With Code. This enables us to build a knowledge graph in which different models are associated with different benchmarks and tasks. Apart from the relevant performance scores in the various benchmarks and the models' properties extracted from HuggingFace, we extract from the model cards coding snippets that describe how each model can be loaded and executed. These execution snippets enable on-demand loading of a chosen model and execution of its inference step based on the provided parameters, if selected by model selection pipelines (see Section 3.2.2).

When processing the HuggingFace model card, we leverage a combination of keywords (see Appendix D) and regular expressions to identify coding blocks that showcase simplified examples of loading and executing a particular model $m_j$. After the coding blocks are extracted, we prompt (see Appendix D) an LLM that is proficient in code generation, such as DeepSeek-Coder[8], to generate a suitable Python function invoking $m_j$ and running inference while taking into consideration input arguments of the originally extracted coding block. The resulting Python function, together with its signature (returned type, variable types and default values), is finally stored within the C-KG and connected with its corresponding model using an `execution` $\in E$ edge.

## 3.2 PLANNING MODEL ACTIONS

Having extracted and systematically structured the information pertinent to models associated with various tasks, we are now positioned to delineate the specific actions required to accomplish the objectives in the user query, as depicted in Figure 1.

---

[4]https://paperswithcode.com/

[5]https://arxiv.org/

[6]arXiv papers may report results of different model variants, *e.g.* performance across different models sizes.

[7]https://huggingface.co/google/flan-t5-base

[8]https://huggingface.co/deepseek-ai/DeepSeek-Coder-V2-Base

### 3.2.1 TASK PLANNER

**Parsing User Query**    User queries are often vague and unstructured, making it challenging for systems to understand the user's intent accurately. To overcome this, we introduce a parsing-rephrasing stage that bridges the gap between the ambiguous query and the system's structured requirements. This initial step sets the foundation for the subsequent stages, enabling the system to extract relevant information.

We parse an input user query $\mathbf{q}$ into distinct components: instruction ($\mathbf{i}$: str), input text ($\mathbf{t}$: str), question ($\mathbf{s}$: str), URL ($\mathbf{u}$: str), data ($\mathbf{x}$: dict), and categories ($\mathbf{g}$: list), as follows:

$$\mathcal{P}(\mathbf{q}) = \{\mathbf{i}, \mathbf{t}, \mathbf{s}, \mathbf{u}, \mathbf{x}, \mathbf{g}\} \tag{1}$$

with $\mathcal{P}$ the parsing function. The ability of LLMs to parse user queries into structured formats, like JSON, has been highlighted across multiple research efforts (Petroni et al., 2019; Wei et al., 2023). Using prompt engineering with a few-shot setting Brown (2020) (see Appendix D), we guide the LLM to convert an unstructured user input into structured data. Additionally, we ask the LLM to rewrite the user instruction to enhance clarity, simplifying complex directives and converting implicit information into explicit statements.

The instruction is crucial as it is used in the later stages to decompose the user query into smaller parts and determine objectives in the user query, $\mathbf{q}$. By transforming vague queries into well-defined components, our system becomes more robust to handle diverse and complex user inputs.

**Task Decomposition**    Given the resulting instruction, after processing the input user query $\mathbf{q}$, we proceed to decompose it into smaller, manageable steps to identify a specific plan to attain the objectives (Wei et al., 2022; Yao et al., 2023b) and understand how each part of the instruction is associated with achievable goals within the system's capabilities. We utilise an LLM as a classifier (Zhang et al., 2024) in a few-shot example setting (Brown, 2020) (see Appendix D) to identify the relevant domains from the original set of PDDL domains, $\mathbb{D}$.

By providing the LLM with examples of instructions and their associated domains, we guide it to select the pertinent subset $\mathbb{D}^* \subseteq \mathbb{D}$ that aligns with the instruction. We prioritise *recall* in this classification step to ensure that all potentially relevant domains are considered, minimizing the risk of missing critical actions required to fulfil the user's objectives. This approach enhances the system's robustness by accounting for a wider range of possible actions.

Once we determine the subset of relevant PDDL domains, $\mathbb{D}^*$, we leverage the predefined PDDL structures of each classified domain. These domain structures are then mergedto create a unified PDDL domain file inclusive of all actions from $\mathbb{D}^*$, s.t.

$$\mathbf{a}^{\mathbb{D}^*} = \bigcup_{j=1}^{|D|} \mathbf{a}^{d_j} \,\Big|\, d_j \in \mathbb{D}^*. \tag{2}$$

This method ensures that the combined domain file encompasses all necessary actions while maintaining consistency and comprehensibility. Next, we exploit the parsed instruction $\mathbf{i}$ and the compiled set of actions from $\mathbb{D}^*$, $a^{\mathbb{D}^*}$, with an LLM to determine the specific actions required to achieve the instruction's objectives (see Appendix D), as follows: $\mathbf{a}_{\mathbf{i}}^{\mathbb{D}^*} \subseteq \mathbf{a}^{\mathbb{D}^*}$.

This allows to precisely map high-level user intents to concrete actions. Following this selection, the combined PDDL domain file (i.e. $\mathbf{a}^{\mathbb{D}^*}$) and the identified actions set (i.e. $\mathbf{a}_{\mathbf{i}}^{\mathbb{D}^*}$) help to reconstruct the corresponding PDDL problem. Finally, a Best First Width Search (Lipovetzky & Geffner, 2017) logical reasoner computes a detailed Plan of Actions ordering the actions $\in \mathbf{a}_{\mathbf{i}}^{\mathbb{D}^*}$ such that the system can execute step-by-step, ensuring coherent execution in line with the user's intent.

This hierarchical approach not only enhances the system's robustness in parsing and understanding diverse and complex user inputs but also guarantees the accuracy and feasibility of generated actionable plans by rigorously structuring them within established domain constraints.

### 3.2.2 MODEL SELECTION

Following the task planning, the next stage is the selection of appropriate models capable of performing the specified actions in the plan. Utilising the information from our C-KG (see Section 3.1), our goal is to identify a model combination $\mathbb{M}^* \subseteq \mathbb{M}$ that would satisfy a set of conditions

$\mathbb{C} = \{c_1, c_2, \ldots, c_{|C|}\}$ imposed by the user while offering guarantees that the selected model combination will be suitable to generate an output $\mathbf{y}$ that addresses the original input query $\mathbf{q}$, s.t.

$$\mathbb{M}^{\star} = \arg\max_{\mathbb{M}} p\left(\mathbf{y}|\mathbf{q}, \mathbb{M}, \mathbb{C}\right). \qquad (3)$$

For instance, if licensing is a concern, we filter out models that do not meet the required licensing terms. Similarly, if there are limitations on computational resources, we prioritise models that are efficient in size and cost. Additionally, we consider performance scores from relevant benchmarks to guide our selection. By analyzing these performance metrics, we can choose models that have demonstrated high effectiveness on tasks similar to those required, ensuring that the selected models are not only compliant with user constraints but also optimal for the specific tasks at hand.

By balancing these factors: model capabilities, licensing requirements, resource constraints, and performance metrics, we systematically select the most suitable model for each action. This ensures that the execution of the plan is aligned with both the technical requirements of the tasks and the practical considerations of the user. Ultimately, this careful selection enhances the system's efficiency and effectiveness, enabling it to perform complex tasks while adhering to user constraints.

### 3.2.3 PLAN EXECUTION

With the appropriate models selected for each action, we proceed to the plan execution phase. The execution involves retrieving the Python code snippets associated with `execute` relation with the chosen models, which are stored in the C-KG. More specifically, we map the arguments of the Python functions to the relevant components extracted from the parsed user query—$\{\mathbf{i}, \mathbf{t}, \mathbf{s}, \mathbf{u}, \mathbf{x}, \mathbf{g}\}$ (cf. Eq. 1) —using a complex similarity mapping algorithm. This mapping ensures that the models receive the correct inputs derived from the user's query, accurately capturing the user's intent. In cases where the Python code snippet for the selected model is not available, we employ a fallback strategy. We search for code snippets from other models that have been assigned to the same task and possess similar functionalities. This approach leverages the semantic and functional similarities between models within the same domain, allowing us to substitute models when necessary without compromising the action's intended outcome. By orchestrating the retrieval of code snippets and the fine-grained mapping of arguments, our system seamlessly transforms high-level plans into executable code, ensuring that the models operate on the intended data and parameters. This execution phase is critical as it bridges the gap between planning and action, which guarantees that each step of the plan is performed correctly and efficiently.

## 4 EXPERIMENTS

### 4.1 BASELINES

Addressing complex, multi-modal real-world tasks presents substantial challenges, and current solutions are limited. For our experiments, we compare our proposed method against the most relevant state-of-the-art techniques: HuggingGPT (Shen et al., 2024a) and ControlLLM (Liu et al., 2023c). These represent significant advancements in integrating LLMs with task planning and execution frameworks.

We propose two innovative methods: HIVE and HIVE light. HIVE leverages the advanced capabilities of ChatGPT for parsing user queries and decomposing tasks. To address the computational challenges associated with ChatGPT-based systems, we designed HIVE light as an efficient alternative that can be deployed on local servers. HIVE light employs InterLM2.5-7B-chat[9] Cai et al. (2024) for parsing user queries and Mistral-7B-Instruct-v0.3[10] for task decomposition, both of which have been subjected to 8-bit quantization. We selected a chat-oriented model for parsing, as conversational models excel at understanding subtle queries. Additionally, an instruction fine-tuned model was chosen for task decomposition due to its capability to deliver precise instruction clarity. By employing this dual-method setup, we ensure a thorough performance evaluation, positioning HIVE and HIVE light as strong competitors to existing state-of-the-art frameworks.

---

[9] https://huggingface.co/internlm/internlm2_5-7b-chat
[10] https://huggingface.co/mistralai/Mistral-7B-Instruct-v0.3

Table 1: Comparison of Task Selection, Flow of Tasks, and Output across all competitors.

| Query Types | HuggingGPT | | | ControlLLM | | | HIVE light | | | HIVE | | |
|---|---|---|---|---|---|---|---|---|---|---|---|---|
| | TS | FoT | O | TS | FoT | O | TS | FoT | O | TS | FoT | O |
| **Single Task** | 0.47 | 0.47 | 0.83 | 0.74 | 0.74 | 0.74 | 0.80 | 0.80 | 0.72 | 0.88 | 0.88 | 0.79 |
| **Two Tasks** | 0.64 | 0.55 | 0.44 | 0.33 | 0.33 | 0.38 | 0.67 | 0.62 | 0.52 | 0.71 | 0.69 | 0.58 |
| **Three Tasks** | 0.42 | 0.42 | 0.30 | 0.36 | 0.36 | 0.33 | 0.57 | 0.43 | 0.33 | 0.67 | 0.67 | 0.46 |
| **Overall** | 0.57 | 0.51 | 0.53 | 0.43 | 0.43 | 0.47 | 0.69 | 0.64 | 0.55 | **0.74** | **0.73** | **0.62** |

## 4.2 MuSE — Multi-modal Sub-task Execution Benchmark

In the absence of standard publicly available benchmarks for solving real-world tasks, and recognizing that HuggingGPT (Shen et al., 2024a) did not release their evaluation dataset, we developed a new benchmark to assess our proposed framework alongside state-of-the-art methods like Hugging-GPT (Shen et al., 2024a) and ControlLLM (Liu et al., 2023c). Although ControlLLM (Liu et al., 2023c) released their benchmark, it utilises a fine-tuned task decomposer, which could introduce bias if used for our evaluation. Therefore, to ensure a fair and unbiased comparison, we collaborated with experts from diverse linguistic backgrounds to create a set of 100 heterogeneous, real-world user queries (see Appendix Table 5) These queries are categorised into three types: Single-task, Two-task, and Three-task queries. In order to facilitate a fair comparison, we included only those tasks and models that are supported by all three systems. The benchmark is designed to cover various task domains, such as automatic speech recognition, question answering, and image generation, involving 15 models across different modalities[11]. This comprehensive benchmark enables us to rigorously evaluate the performance and generalisability of our framework.

**Metrics.** To assess our framework against state-of-the-art methods, we evaluate performance on three fronts, using binary metrics for simplicity and clarity:

- **Task Selection (TS)**: Determines whether the system accurately identifies the required tasks from the user's query. We assign a binary score of 1 if the system selects all the tasks correctly, and 0 if it does not or if it selected irrelevant tasks. Correct task selection is crucial as it lays the foundation for successful execution and directly impacts the relevance of the final output.

- **Flow of Thought (FoT)**: We evaluate the logical sequence and integration of the selected tasks. A binary score is given based on whether the system establishes the correct flow—1 for a proper flow that respects task dependencies and order, and 0 for an incorrect sequence. This ensures that, especially in multi-task queries involving two or three tasks, the system processes tasks in an order that leads to the desired outcome.

- **Final Output (O)**: Assesses the correctness of the system's final response to the user's query. We adopt a binary evaluation—1 if the output fulfills the user's requirements, and 0 if it falls short. This includes evaluating the relevance of generated content, and the overall satisfaction of the user's intent. Note that we do not evaluate the *quality* of the output, that is we do not judge if the output is accurate but only focus on whether the expected task has been performed.

By employing these binary metrics for each query, we simplify the evaluation process while effectively capturing the essential aspects of each system's performance.

## 4.3 Overall Results

The results of the experiment are presented in Table 1. The scores highlight the effectiveness of our proposed approach in handling complex multi-modal tasks. HIVE consistently outperforms the baseline methods in overall performance across all evaluation metrics: TS, FoT, and O. This performance is evident in both single-task and multi-task scenarios. Remarkably, HIVE light, which

---

[11]MuSE involves **10** domains leading to 10 distinct PDDL domain files covering **15** tasks. 67% of the queries are multi-modal [text,image,audio], see also Appendix A and the supplementary material.

Table 2: Cross-Modality Performances.

| In↓   Out→ | Text | | |
|---|---|---|---|
| | **HuggingGPT** | **ControlLLM** | **HIVE light** |
| **Image** | 0.48 | 0.45 | **0.52** |
| **Audio** | 0.25 | 0.25 | **0.67** |

| In↓   Out→ | Image | | |
|---|---|---|---|
| | **HuggingGPT** | **ControlLLM** | **HIVE light** |
| **Text** | 0.36 | 0.18 | **0.75** |
| **Audio** | 0.50 | 0.25 | **1.00** |

| In↓   Out→ | Audio | | |
|---|---|---|---|
| | **HuggingGPT** | **ControlLLM** | **HIVE light** |
| **Text** | 0.33 | 0.14 | **0.71** |
| **Image** | **0.80** | 0.00 | 0.00 |

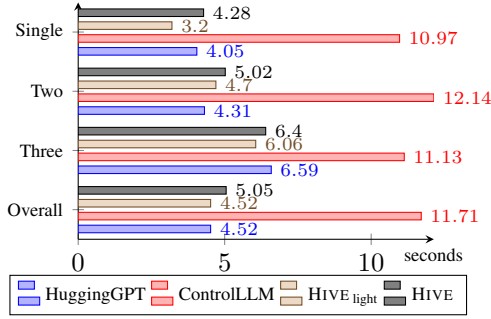

Figure 2: Average times (s) before execution.

is based on 8-bit quantised models with only 7 billion parameters, surpasses both HuggingGPT and ControlLLM in the overall evaluation. This underscores the effectiveness of our approach even with reduced computational resources.

In the case of single-task queries, all methods perform relatively well in generating correct final outputs. Though, HIVE stands out by achieving the highest performance in Task Selection and Flow of Thought, indicating a more accurate and coherent understanding and reasoning process. Conversely, HuggingGPT attains the highest Final Output performance but performs poorly in Task Selection and Flow of Thought metrics. This discrepancy arises because HuggingGPT tends to collect as many relevant tasks as possible, even if a query requires only a single task, leading to over-selection. Despite this over-selection, its strong ChatGPT backbone enables it to produce high-quality final outputs. Nevertheless, this approach may reduce the trustworthiness of the results, as it does not align precisely with the intended tasks (further discussed in Section 4.4.2). When it comes to multi-task queries, as expected, HIVE distinctly outperforms its competitors across all metrics. PDDL planning in HIVE light allows it to surpass HuggingGPT and ControlLLM, demonstrating proficiency in task selection, task flow management and execution. In contrast, ControlLLM struggles considerably with multi-task queries, exhibiting the weakest performance among the evaluated methods. While HIVE, HIVE light, and HuggingGPT employ prompt-based strategies to provide flexibility and adaptability, ControlLLM relies on a fine-tuned task decomposer. This approach limits its capacity to generalise to queries that deviate even slightly from its training set, leading to significant performance declines in multi-task scenarios.

## 4.4 DISCUSSIONS

### 4.4.1 CROSS-MODALITY PERFORMANCES

To gain a better understanding of the multi-modality of these systems' capabilities, we dig deeper into the Final Output (O) results from Table 1. We divide this investigation into three distinct parts based on the output modality and analyze the performance when the other two modalities are involved in the input (Table 2).

When it comes to text output, HIVE light demonstrates a substantial lead over its competitors when the input includes any image or audio. This showcases HIVE light's ability to integrate visual and auditory data to enhance text outputs. In the context of image output, HIVE light once again outperforms the other systems, illustrating its proficiency in converting textual and audio inputs into coherent visual responses. Lastly, although our system shows commendable performance in text-based audio generation, it falls short of achieving the desired objective in the image-to-audio scenario, indicating an area for potential improvement in future iterations.

### 4.4.2 TRUSTWORTHINESS

In order to review the connection between *justifications* (*i.e.* the conjunction[12] of TS and FoT) and *outputs* (O), we group in Figure 3 the results based on (justification, output) scores which can

---

[12]⊤ iff TS AND FoT are both 1; ⊥ in all other cases.

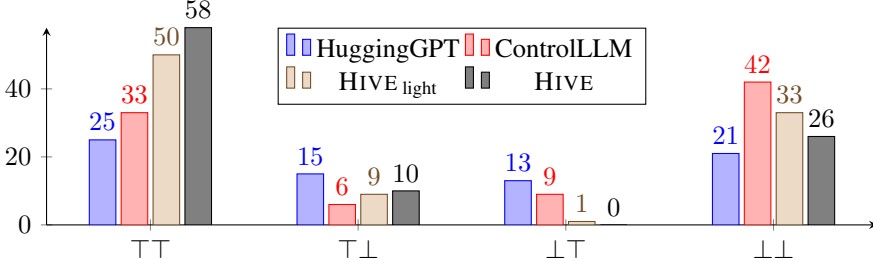

Figure 3: Correlations between *justifications* (TS AND FoT) and *outputs* (O).

respectively be correct ⊤ or incorrect ⊥. The failure cases `Err` are further discussed in Section 4.4.3. There are thereby four distinct cases. First, ⊤⊤ which corresponds to fully correct cases having both justifications and outputs, in this HIVE solutions (50+) outperform both HuggingGPT (25) and ControlLLM (33). Then, ⊤⊥ means that the plans were correct but the execution did not go through. In this category the four reviewed systems perform similarly, ranging from 6 to 15 cases. On the right-end side of Figure 3, the plain-wrong case ⊥⊥ witnesses ControlLLM as the "worst" system (see results in Section 4.3), having the highest score. Finally, the critical ⊥⊤ case indicates a lack of trustworthiness across all the baseline systems that may result in misleading outcomes. This case involves an incorrect plan or justification, despite the output being correct. Such cases have been recorded 9 times for ControlLLM and 13 times for HuggingGPT while being absent from HIVE and singleton in HIVE light showcasing the reliability of the results produced by HIVE.

Overall, this discussion allowed us to highlight two aspects. One, unlike other solutions, HuggingGPT exhibits results ranging from 13 to 25 in the four categories, meaning that it is hard to rely on it. Second, HIVE (s) tend not to fall in incoherent cases where results are correct without the corresponding plan being correct—in other words, when results are good their explanations can be trusted as well.

### 4.4.3 ROBUSTNESS

Since most of MUSE's queries require multiple models to interact together in a compatible manner, we noticed that sometimes the tested systems fail at dealing with either the *justification* or the *output* parts. In Table 3, we list the different cases encountered. The first point to be highlighted is that, among the four systems, HuggingGPT is the less robust one by far: 22 `Err` against 10, 7 and 6 for the other solutions. Second, even more critical, is that HuggingGPT, unlike the competition, is able to generate correct results (⊤) while failing (`Err`) in its plan construction. This exacerbates the fact that its justifications cannot be trusted, as the executed actions tend in many cases not to be consistent with the compiled plan, using GPT-3.5 at most places. This last finding is coherent with the ⊥⊤ discussion in Section 4.4.2.

### 4.4.4 LATENCY FOR PLANNING

Lastly, in this *discussion* section, we analyse the time performances (in seconds) of the systems to come up with a plan and select suitable models. As the chosen models may differ and no enforced rules such as "the quicker the better" (see Section 4.5 for discussion about model selection capabilities) were added, we measure the latencies up to the model selection stage.

Figure 2 presents these latencies according to the split already presented in Section 4.3 as per the number of tasks involved in the queries. First of all, HuggingGPT and HIVE (s) share the same orders of magnitude whereas ControlLLM is a magnitude slower (always 10+ seconds). Next, as expected, the more tasks within the query the slower the systems become until they reach an execution plan. On this, it is interesting to note that HuggingGPT's scaling law does not seem "linear" as the slope increases greatly between two- and three-task queries. This behaviour is compatible with the internal implementation of HuggingGPT: when other systems are rule-based (see the C-KG for HIVE (s) to select models), HuggingGPT needs to prompt (together with model descriptions) to select which models to use for each task.

Table 3: Failing case enumeration (Err), either as *justifications* (TS AND FoT) or *outputs* (O).

| Failure Type | HuggingGPT | ControlLLM | HIVE light | HIVE |
|:---:|:---:|:---:|:---:|:---:|
| Err, ⊤ | 8 | 0 | 0 | 0 |
| ⊤, Err | 3 | 1 | 3 | 3 |
| ⊥, Err | 3 | 1 | 1 | 0 |
| Err, Err | 8 | 8 | 3 | 3 |
| **Overall** | 22 | 10 | 7 | 6 |

## 4.5 TAKING INTO ACCOUNT USERS' CONSTRAINTS IN TERMS OF MODEL SELECTION

As depicted in Section 3.2.2, once a plan of actions is established, HIVE selects the *best* models to realise them. Obviously, depending on the circumstances, the definition of what is "best" may vary a lot, *e.g.* when resources are sparse, one may decide to use the smallest models possible even if the resulting quality is reduced, alternatively users might choose to select models based on their respective (recorded) results for specific benchmarks. In order to respect these various cases, HIVE allows users to specify selection criteria. In this Section[13], we review the capabilities of HIVE against HuggingGPT when users want to force some conditions of their own in the model selection. Since ControlLLM has one-to-one mappings of models for each task, it is *de facto* excluded.

Practically, we use the following query[14]: "`Transcribe the audio from .audio_1.wav and find entity tokens`". Regarding the task-model mappings, we let both HIVE and HuggingGPT have access to: openai/whisper-large-v2 and nvidia/parakeet-rnnt-1.1b (having respectively Apache-2.0 and CC-By-4.0 for licenses) for the ASR; and to dslim/bert-base-NER (MIT license) for NER. We first run the query without any constraints (control run, see Appendix B): both systems, HIVE and HuggingGPT, were able to transcribe the audio file and perform NER (even though HuggingGPT result set was empty). We then applied the following model selection constraints sequentially:

1. **License restrictions**: only use Openrail++ and Deepseek — HIVE returned nothing which was the expected behaviour as the available models were not having the requested licenses; on the other hand, HuggingGPT performed the task as in the control therefore infringing the restrictions (see Appendix B).

2. Uses the "**smallest possible**" model[15] — HIVE complied with the user choice and used the smaller models whereas HuggingGPT kept using openai/whisper-large-v2 as in the control run (see Appendix B).

3. Filter for the model having the **best results** at the `speech recognition on common voice english`[16] benchmark — Using the benchmark records from the C-KG, HIVE was able to select the correct model unlike HuggingGPT which chose models like in the control run (see Appendix Table B).

Overall, HIVE answered each time while properly taking into account the given constraints. While HuggingGPT failed every time, misleading even the users with regards to its justifications (refer to Section 4.4.2 for further justifications on this).

## 5 RELATED WORK

**Automated Planning.** The cognitive ability to organize and coordinate actions toward a specific goal is referred to as *planning*. While humans innately possess this capacity, machines lack such a capability. Automated planning has garnered significant interest from researchers across various domains, including robotics (Guo et al., 2023), autonomous vehicles (Madridano et al., 2021), and dialogue systems (Wang et al., 2023a). The methodologies employed to devise sequences of actions have evolved considerably, particularly in light of recent breakthroughs in deep learning. Before the

---

[13] See also Appendix B for an extensive result description.

[14] A multimodal one involving two tasks: ASR and NER.

[15] We use the model disk footprint as a proxy for its size.

[16] Introduced in "Common Voice: A Massively-Multilingual Speech Corpus" Ardila et al. (2019).

advent of large language models (LLMs), planning frameworks such as STRIPS (Fikes & Nilsson, 1971) or HTN (Erol et al., 1994) were developed to decompose tasks into a series of actions (or sub-tasks) leading to the desired outcomes (Sacerdoti, 1975). Building upon these frameworks, the Planning Domain Definition Language (PDDL) (Aeronautiques et al., 1998) emerged as a widely adopted standardized language for defining planning problems and domains. However, LLMs have superseded those frameworks to stand as a planner on their own (i.e. the *LLM-as-planner* paradigm). Multiple prompt engineering techniques (Liu et al., 2023b; Graux et al., 2024) were designed to leverage in-context learning aiming to directly generate the multi-step problem solutions. More specifically, the Chain-of-Thought (Wei et al., 2022) has revealed the promising reasoning capabilities of LLMs, and therefore new techniques were fashioned such as the self-consistency decoding strategy (Wang et al., 2022), Tree-of-Thought (Yao et al., 2023a), Program-of-Thought (Chen et al., 2023) or Graph-of-Thought (Yao et al., 2023c; Besta et al., 2024). However, LLMs are still struggling to produce acceptable and logical plans, especially as the complexity of the problem increases (Valmeekam et al., 2023; Xiao et al., 2024; Zheng et al., 2024) Thus, numerous initiatives have therefore sought to integrate problem-specific languages like PDDL along LLMs to maximize their effectiveness and leverage their full potential (Vyas et al., 2025; Pallagani et al., 2023; Liu et al., 2023a; Oswald et al., 2024).

**LLM-as-Agent.**   The genesis of large language models (LLMs) primarily stemmed from textual content, which initially narrowed the research focus to text generation. However, to address the diversity of real-world scenarios, significant efforts have been directed toward developing vision or speech LLMs, thereby aligning with a multi-modal paradigm (Zhu et al., 2023; Wu et al., 2023; Wang et al., 2023b). Additionally, to expand the capabilities of LLMs, there has been an increasing trend to integrate them with external tools incorporating retrieval (Shen et al., 2024c;b) or other tool learning strategies (Wang et al., 2024) Toolformer (Schick et al., 2024) pioneered the invocation of tool calls within generated sequences via special tokens giving rise to tool-augmented LLMs (Qin et al., 2023a;b; Guo et al., 2024; Qu et al., 2024). Then, ReAct (Yao et al., 2023b) introduced such intermediate tool calls during the reasoning process by incorporating intermediate outcomes within the prompt to better guide the final resolution of the problem. In contrast to ReAct, Reflexion (Shinn et al., 2023) adds verbal feedback on those intermediate results to further assess and verify outcomes, In the meantime, a plethora of fine-tuned LLMs tailored for specific tasks has become ubiquitous on platforms such as Hugging Face Hub (Wolf et al., 2019), alongside proprietary models such as GPT-4 (Achiam et al., 2023), Claude (Anthropic, 2023), and Gemini (Team et al., 2023) offering the opportunities to consider these LLMs as distinct agents. Indeed, the gathering of technical details for each parametric model stands as a critical component in the reporting and tracking efforts underlined by the use of *Model Cards* Mitchell et al. (2019). HuggingGPT (Shen et al., 2024a), leverages such a large pool of LLMs using ChatGPT as the core controller. Following a similar approach, ControlLLM (Liu et al., 2023c) and Chameleon Lu et al. (2024) explore task planning via prompt engineering and integrate a more diverse pool of tools. While HuggingGPT, ControlLLM or Chameleon appoint appropriate models for each sub-task, however, their model selection process remains sub-optimal as they do not identify the most accurate model. Thus, if these frameworks can fulfil their plans, the resulting performance may be unsatisfactory if the best agent is not utilized. To the best of our knowledge, our work represents the first attempt to address this gap.

## 6   CONCLUSION

Our research introduces HIVE, an innovative and comprehensive knowledge-aware planning solution designed to navigate the complexities of model selection and task planning using a diverse set of deep learning models. By leveraging a Capability Knowledge Graph and an LLM-based formal logic planner, we transcend the limitations of the existing systems. HIVE stands out for its capability to plan and explain complex action chains while respecting user-specific constraints –thereby achieving both high performance and full transparency. Empirical evaluations on our newly designed benchmark reveal HIVE's superior performance, consistently outperforming competing platforms like HuggingGPT and ControlLLM. This breakthrough underscores HIVE's potential to redefine the state-of-the-art in task selection and planning, ultimately facilitating more efficient and user-friendly applications of advanced deep models. HIVE thus advances the handling of multi-modal tasks.

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

## A    MUSE EXPERIMENTAL SETUP

Table 4 provides a comprehensive overview of the domains and tasks encompassed within the MUSE benchmark. All three competing systems have access to the models associated with each task listed in Table 4. It should be noted that both HuggingGPT and ControlLLM utilise ChatGPT as their backbone model, leveraging it for the execution of certain tasks. Since ControlLLM permits only one-to-one mappings, to maintain an unbiased benchmark, we assign one model per task (for details, please refer to the `Supplementary Material` repository[17]). Moreover, to preserve the naturalness of the queries, we have refrained from making any grammatical or spelling corrections in the dataset.

Table 4: AI Tasks and Associated Models.

| Domain | Task | Model |
|---|---|---|
| Audio | Automatic Speech Recognition
Text to Speech | openai/whisper-large-v2 (Radford et al., 2022)
microsoft/speecht5_tts (Ao et al., 2022) |
| Image Generation | Text to Image | stabilityai/stable-diffusion-2-1(Rombach et al., 2022) |
| Image to Text | Image Captioning
Object Detection
Visual Question Answering | Salesforce/blip-image-captioning-base(Li et al., 2022b)
facebook/detr-resnet-101(Carion et al., 2020)
Salesforce/blip-vqa-base(Li et al., 2022a) |
| Image to Image | Depth Estimation | Intel/dpt-hybrid-midas(Ranftl et al., 2021) |
| Machine Translation | Translation from xx to yy | mistralai/Mistral-7B-Instruct-v0.1(Jiang et al., 2023) |
| Question Answering | Answer based on Context
Open QA
Table QA | distilbert/distilbert-base-uncased-distilled-squad(Sanh et al., 2019)
mistralai/Mistral-7B-Instruct-v0.1(Jiang et al., 2023)
Yale-LILY/reastap-large(Zhao et al., 2022) |
| Summarisation | Abstractive Summarisation | facebook/bart-large-cnn(Lewis et al., 2019b) |
| Text Generation | Text Generation | mistralai/Mistral-7B-Instruct-v0.1(Jiang et al., 2023) |
| Text Classification | Text Classification | facebook/bart-large-mnli (Lewis et al., 2019a) |
| Token Classification | Named Entity Recognition | dslim/bert-base-NER(Devlin et al., 2018) |

Table 5: Small sample of queries from the MUSE benchmark.

| Domain | Queries |
|---|---|
| "object_detection" | How many people is in the image? ./data/images/image_6.jpg |
| "image_to_text", "text_generation" | Write me a sonnet about this image ./data/images/image_11.jpg |
| "image_to_text", "image_generation" | Build a pictorial presentation of the flower after deciphering the audio ./data/audios/audio_4.wav |
| "audio", "question_answering" | Use audio transcription as context and answer 'what is the country starting the war whom the associated speech is about?' from ./data/audios/audio_6.wav |
| "audio", "summarisation", "image_generation" | Extract a summary from the audio transcript ./data/audios/audio_8.wav, and generate an image based on the summary |

---

[17]https://github.com/dgraux/Hive-ICLR-2025

## B  SCENARIOS DETAILS FOR MODEL SELECTION VALIDATION

⇊ ⇊    **No Constraints (Cold run)**    ⇊ ⇊

### HIVE light

We, the people of the United States, in order to form a more perfect Union, establish justice, ensure domestic tranquility, provide for the common defense, promote the general welfare, and secure the blessings of liberty to ourselves and our posterity, do ordain and establish this Constitution for the United States of America.

```
{
    "LOC": [
        "United_States",
        "United_States_of_America"
    ],
    "MISC": [
        "Constitution"
    ]
}
```

**Comment**: *Successfull!*

### HuggingGPT

Based on the inference results, I have transcribed the audio file ".[Audio]" and obtained the following text: "We, the people of the United States, in order to form a more perfect union, establish justice, ensure domestic tranquility, provide for the common defense, promote the general welfare, and secure the blessings of liberty to ourselves United States of America."

For the named entity recognition task, I have used the model "dslim/bert-base-NER." However, the inference result for this task is empty, indicating that no named entity tokens were found in the transcribed text.

Here is the complete workflow for your request:

1. **Automatic Speech Recognition (ASR) Task:**
   **Model used:** "openai/whisper-large-v2"
   **Inference result:** The audio file ".[Audio]" was transcribed, and the resulting text is the one mentioned above.

2. **Named Entity Recognition (NER) Task:**
   **Model used:** "dslim/bert-base-NER"
   **Inference result:** No named entity tokens were found in the transcribed text.

Please note that the inference results are based on the models used, and there is a possibility of errors or incomplete detection.

**Comment**: *Unsuccessful: the system failed to pass the output from the previous task as input to the subsequent one, preventing the NER model from identifying any entities*

⇊ ⇊    **Specifically asked to use the smallest model available for each task**    ⇊ ⇊

### HIVE light

We, the people of the United States, in order to form a more perfect Union, establish justice, ensure domestic tranquility, provide for the common defense, promote the general welfare, and secure the blessings of liberty to ourselves and our posterity, do ordain and establish this Constitution for the United States of America.

```
{
    "LOC": [
        "United_States",
        "United_States_of_America"
    ],
    "MISC": [
        "Constitution"
    ]
}
```

**Comment**: *Successful!*

### HuggingGPT

Based on the inference results, I have transcribed the audio file ".[Audio]" and found the following named entity tokens:
"We the people of Unites States" – ORGANISATION
"the United States of America" – LOCATION
To accomplish this, I followed the following workflow:

1. **Task 1: Audio-to-Audio Transcription Model:** No suitable model found for this task
   **Inference Result Error:** "audio-to-audio" not found in available tasks

2. **Task 2: Automatic Speech Recognition (ASR) Task:**
   **Model:** I used the "openai/whisper-large-v2" model
   **Inference result:** "We, the people of the United States, in order to form a more perfect union, establish justice, ensure domestic tranquility, provide for the common defense, promote the general welfare, and secure the blessings of liberty to ourselves United States of America."

3. **Task 3: Named Entity Recognition (NER) Task:**
   **Model:** I used the "dslim/bert-base-NER" model
   **Inference result:** No named entity tokens were found in the transcribed text.

I apologize for not being able to provide any named entity tokens in the transcribed text. If you have any further questions or need assistance with anything else, please let me know!

**Comment**: *Unsuccessful: the system failed in task selection and selected "audio-to-audio" as the first task, which is irrelevant to the user query. The system also failed to identify any entities*

⇓⇓ **Specifically ask the systems to use `openrail++` & `deepseek` licensed models** ⇓⇓

**HIVE** light

---

**Comment**: *Successful: as there were no models with `openrail++` and `deepseek` for ASR or NER, no model should be selected*

**HuggingGPT**

I apologize for any confusion caused. Based on the inference results, here is the detailed workflow for your request:

1. **Automatic Speech Recognition (ASR) Task:**

   **Task:** Transcribing the audio file "[Audio]".
   **Model used:** "openai/whisper-large-v2"
   **Inference result:** The transcribed text is as follows: "We, the people of the United States, in order to form a more perfect union, establish justice, ensure domestic tranquility, provide for the common defense, promote the general welfare and secure the blessings of liberty to ourselves and our posterity, do ordain and establish this Constitution for the United States of America."

2. **Named Entity Recognition (NER) Task:**

   **Task:** Finding named entity tokens in the transcribed text.
   **Model used:** "dslim/bert-base-NER".
   **Inference result:** Unfortunately, there is no available inference result for this task.

3. **Zero-shot Classification Task: Task:** Classifying named entity tokens.
   **Model used:** "gpt-3.5-turbo-0613"
   **Inference result:** I apologize, but as an AI language model, I am unable to access or transcribe audio directly. However, once the audio is transcribed, I can assist you with the zero-shot classification task. Could you please provide the transcription of the audio file, and I'll be happy to help you with the classification task using the provided text and image information?

4. **Text-to-Image Task: Task:** Generating images based on named entity tokens.
   **Model used:** "stability/stable-diffusion-v1"
   **Inference result:** Unfortunately, there was an error in the inference process for this task.

Please note that the inference results are based on the available models and their capabilities. If there are any specific requirements or modifications you would like to make, please let me know.

---

**Comment**: *Unsuccessful: first the system performed an incorrect selections of tasks and then also misguided the user by selecting models that do not aligned with the users license contraints*

⇓⇓ **Specifically asked to choose the Automatic speech recognition model based on best performance on `speech recognition on common voice english`** ⇓⇓

**HIVE** light

we the people of the united states in order to form a more perfect union establish justice insure domestic tranquillity provide for the common defense promote the general welfare and secure the blessings of liberty to ourselves and our posterity do ordain and establish this constitution for the united states of america

```
{
    "LOC": [
        "united_States",
        "united_States_of_
            ↪ america"
    ],
    "MISC": [
        "constitution"
    ]
}
```

---

**Comment**: *Successful: to test the model selection on benchmark performances, we added a "parakeet-rnnt" model in the mix as on this specific benchmark it performs better than the "whisper-large-v2". HIVE light successfully selects this for the ASR task*

**HuggingGPT**

Based on the Inference results, I will directly answer your request and provide a detailed workflow for your specific task. Here is the complete path or URLs of the files in the inference results:

1. **Transcribing the audio file "[Audio]" Model used:** "openai/whisper-large-v2"
   **Inference Result:** The transcribed text is as follows: "We, the people of the United States, in order to form a more perfect union, establish justice, ensure domestic tranquility, provide for the common defense, promote the general welfare, and secure the blessings of liberty to ourselves United States of America."

2. **Finding named entity tokens in the transcribed text:**
   **Model used:** "dslim/bert-base-NER"
   **Inference result:** No named entity tokens were found in the transcribed text.

3. **Zero-shot classification for the task of "named entity tokens":**
   **Model used:** "gpt-3.5-turbo-0613"
   **Inference result:** As an AI language model, I don't have direct access to external files or the ability to process audio. Therefore, I am unable to perform zero-shot classification on the named entity tokens.

4. **Generating a response for the task of selecting models based on scores from "speech-recognition-on-common-voice-english":**
   **Model used:** "gpt-3.5-turbo-0613"
   **Inference result:** Based on the input the following response is generated:
   "To select models for speech recognition on Common Voice English, you can use the scores obtained from the speech recognition systems trained on the Common Voice English dataset. By evaluating the performance of different models using these scores, you can determine which models are most suitable for the task. This allows you to choose the best models for your specific speech recognition needs. Factors such as accuracy, robustness and language coverage should be considered when selecting these models. THe scores provide a valuable metric for comparing and evaluating different models, ultimately assisting in the decision-making process."

Please not that the information provided is based on the inference results from the models used. If you have any specific concerns or further questions, feel free to let me know!

---

**Comment**: *Unsuccessful: the system fails at the task selection state while also ignoring the requirements to selected model for the ASR based on the benchmark performance*

## C  HIVE COMPARED TO CURRENT TOP MONOLITHIC LLMS

The recent advancements in models have shown that foundational model capabilities progressed a lot over the past years. This surge could lead to believe that at some point in the future only having one-single-large model to-do-everything would be enough. Such a hope, has two issues:

1. such a powerful model does not exist yet, and no one knows when it could arise;

2. it is barely impossible that such a model could exist and cover all the "niche" use cases and tasks which are currently populating the model landscape (*e.g.* protein folding or other rare scenarios).

To go further with the first aforementioned point, we ran MUSE over variants of some of the best models currently available[18]: o1-preview and GPT-4o both from OpenAI and DeepSeek-v3 from the eponymous company. The results are as follows:

Since o1-preview does not support image and audio modality, we only test the cases whose input, output and any expected intermediate result only contains text. Overall, o1-preview was able to properly answer (O) **24%** of the time and properly justify (TS & FoT) its choices **26%** of the time.

Table 6: o1-preview.

| Domains | FS | FoT | O | Support |
|---|---|---|---|---|
| One | 11 | 11 | 10 | 12 |
| Two | 10 | 10 | 11 | 12 |
| Three | 5 | 5 | 3 | 5 |
| **Total** | 26 | 26 | 24 | 29 |

Table 7: GPT-4o.

| Domains | FS | FoT | O | Support |
|---|---|---|---|---|
| One | 16 | 16 | 15 | 20 |
| Two | 24 | 24 | 23 | 24 |
| Three | 4 | 4 | 5 | 7 |
| **Total** | 44 | 44 | 43 | 51 |

Once again, since GPT-4o only supports text and image input and textual output, we filtered out all the entries from MUSE involving unsupported modalities. Overall, GPT-4o was able to properly answer (O) **43%** of the time and properly justify (TS & FoT) its choices **44%** of the time.

Since DeepSeek-v3 handles modalities similarly to GPT-4o, we filtered MUSE the same way. After running, it exhibits good planning performances (45% and 47% for FS and FoT respectively) but as compared to OpenAI tested models, output quality felt down to 25%.

Table 8: DeepSeek-v3.

| Domains | FS | FoT | O | Support |
|---|---|---|---|---|
| One | 19 | 20 | 11 | 20 |
| Two | 20 | 21 | 11 | 24 |
| Three | 6 | 6 | 3 | 7 |
| **Total** | 45 | 47 | 25 | 51 |

Unsurprisingly, even if all models are able to plan and to perform over MUSE for some of its queries, they are not yet able to deal with the richness of real-world multi-modal scenarios as depicted in MUSE.

More generally, when it comes to the second point about the "niche" tasks, the essence of the Capability-KG lies in the fact that its richness allows to retrieve candidate models for a large set of tasks (more than 100 at the moment) and does not limit users to envision instructions revolving around a handful of popular tasks.

Finally, we would like to mention a direction we are currently exploring. So far, the presented C-KG gathers information related to models, but structurally, nothing prevents us from collecting and structuring information related to other objects. For instance, the graph nature of the C-KG led us starting the exploration of having sub-nodes for adapters attached to main-principal-models; these adapter-nodes still carry the same type of information as the main nodes currently listed in the C-KG, gathering performance information for instance. In the future, we think this addition would also allow building complex instruction pipelines which would rely mainly on a single backbone model and this could be a solution for some specific actors having access to only one model.

More generally, we would wrap up emphasising on the fact that even if the big models available at the moment aren't yet capable like a group of selected specialised models, the overall technical architecture of HIVE is still appropriate to tackle scenarios where only one model and associated variants (through adapters for instance) is available.

---

[18]In this setting, both planning and execution parts are performed by the tested monolithic models, which slightly differs from the HIVE setting which delegates the execution to specialized models.

## D   HIVE'S PROMPTS

---

**Usage Snippet Extraction**                                                    **(§3.1)**

You are a python programming expert, mainly used to convert python code snippets into python functions. Follow the rules:
1. You also make sure all the function variables have default value.
2. There should always be input variable with default value that takes the input for the model.
3. Take Model path as a variable with default value of model name
4. Return the model response in the python function

- - - - - - - - - - - - - - - - - - - - - - - - - - - - - - - - - - - - - - - - - - - - -

Transform the code snippets in the text into one signed Python function including all the potential variables and default values for all. Only respond with code and in markdown format "'python"'. {code}

---

**Parsing & Rephrasing**                                                         **(§3.2)**

**Task Decomposition stage:** The AI assistant can parse user input into multiple inputs and fill the relevant keys in the following `JSON`.

```
{"instruction": None, "input_text": None, "question": None,
    ↪ "url": None, "data_dict": {}, "categories": []}
```

**Example 1**
**User:** What is the date mentioned in this audio www.google.com/audio_file.mp3?
**Response:** {{"instruction": "Convert the audio to text and then answer the question", "url": "www.google.com/audio_file.mp3", "input_text": "What is the date?"}}
. . .

. . .

**Example 5**
**User:** Please transcribe the voice into text ./audio/audio_1.mp3 and classify the transcribed text into categories such as 'movie', 'music', 'painting', or 'Other'.
**Response:** {"instruction": "Convert the audio to text, and perform text classification", "url": "./audio/audio_1.mp3", "categories": ['movie', 'music', 'painting', 'Other']}

- - - - - - - - - - - - - - - - - - - - - - - - - - - - - - - - - - - - - - - - - - - - -

Based on the above example, parse the following:
**User:** {USER_INPUT}
**Response:**

Only return a `JSON`
The `JSON` keys are defined below:
instruction: textitwhat are the tasks asked in the question, it maybe one, two or three tasks. Try to find the implicit tasks as well
input_text: extract the original text or context in the input. Do not generate on your own
question: extract if there is any question asked
url: extract url passed in the user query
data_dict: extract dictionary passed in the user query
categories: extract ALL categories mentioned in the user query

Do not generate anything other than a parsable `JSON`

---

---

**Domain Classification** (§3.2)

You are professional in natural language processing task. Find which domains are related with the provided user query? You should pick domains from the following list {domains}. You MUST NOT output other domains not in the provdied list. Here are the examples:

**Example 1:** Answer the following questions in detail and give me a summarisation for the answer
**Domains:** question_answering; summarisation
...
**Example 11:** Summarise the transcript of the audio and find entities in it
**Domains:** audio; summarisation; token_classification

- - - - - - - - - - - - - - - - - - - - - - - - - - - - - - - - - - - - - - - - - - - -

**Provided Query:** {query}. 'The order matters'
**Domains:**

---

**Action Selection** (§3.2)

You are an action selector. Given a **Task** and a list of **Actions**,
Select the crucial/mandatory actions. Follow the instructions below:
1. Pick the least amount of actions that can do the job in **Task**
2. Only select actions that are REQUIRED and NECESSARY for **Task**
3. Focus on precision of selection
4. Do not select EVALUATION or SCORE actions unless explicitly asked for in **Task**

**Example 1**
**Task:** I want to select a schema and then generate a SQL query for the a question
**Actions:** ["Schema-Selection", "generate_SQL", "execute_query", "validate_SQL"]
Can you understand the requirements of the Task and select necessary actions from the Actions. Do not give any explanations, only return a list and nothing else. Select at most three diverse, yet relevant actions.
**Selected_Actions:** ["Schema-Selection", "generate_SQL"]
...
...
**Example 3**
**Task:** Retrieve documents on renewable energy advancements and summarise the latest technologies
**Actions:** ["query_based_summarization", "rank_documents", "retrieve_most_relevant_document", "keyphrase_extraction", "summarization_evaluation", "get_extractive_summarization", "get_abstractive_summarization", "retrieve_multiple_documents"]
Can you understand the requirements of the Task and select necessary actions from the Actions. Do not give any explanations, only return a list and nothing else. Select at most three diverse, yet relevant actions.
**Selected_Actions:** ["retrieve_multiple_documents", "get_extractive_summarization"]

- - - - - - - - - - - - - - - - - - - - - - - - - - - - - - - - - - - - - - - - - - - -

**Task:** {user_instruction}
**Actions:** {actions}
Can you understand the requirements of the Task and select necessary actions from the Actions. Do not give any explanations, only return a list and nothing else. Select at most three diverse, yet relevant actions.
**Selected_Actions:**

---

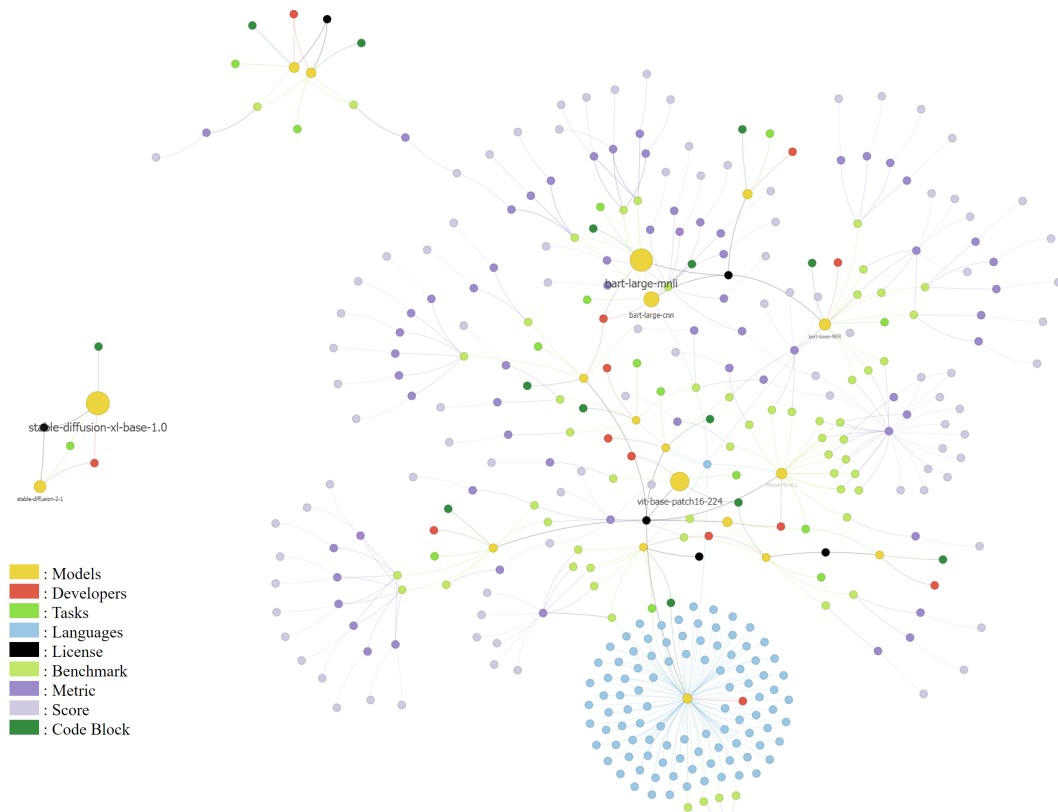

Figure 4: Capability Knowledge Graph in use for the MUSE Benchmark.

# E   MUSE CAPABILITY-KG VISUALISATION

As described in Section 4.2, we introduced the MUSE benchmark in order to compare the performances of HuggingGPT, ControlLLM and HIVE. The latter was declined in two sub-versions: HIVE_light having an 8-bit quantised 7B model for planning tasks and HIVE relying on GPT-3.5 for the same action. In particular, MUSE comes with 100 multi-modal, multi-task, complex, natural-language queries[19].

In this Appendix, we provide, in Figure 4, a snapshot of the Capability Knowledge Graph which corresponds to the models and their associated pieces of data used by HIVE to tackle the MUSE benchmark[20]. It is an excerpt, containing 461 triples involving 381 entities, of the complete C-KG which contains 125k triples for 39k distinct entities. The complete C-KG was formulated by extracting metadata from over 600,000 models listed on HuggingFace, with 26,806 models retained based on popularity metrics. In Figure 4, model nodes are depicted in yellow and are at the core of the knowledge graph, in the sense that it is around models that the C-KG building process was designed, starting from the HuggingFace model cards (see Section 3.1 for more details on this process).

Visually, this graph is twofold, indeed the two stable-diffusion models (xl-base-1.0 & 2-1) are disconnected from the rest of the graph. The remaining 13 models, on the other hand, are making a single piece of graph and organisation nodes (in red) or license nodes (in black) are often connection hubs. Interestingly, one could see that the main part of the C-KG is "bordered" by gray and purple nodes which corresponds to metrics and respective scores for the various benchmark (in bright green) data points retrieved from the `paperswithcode` API. Finally, the *image-to-text* block composed of blip-image-captioning-base and blip-vqa-base is somewhat separated from the main graph and the whisper-large-v2 which is completely surrounded by all the languages it covers (light blue nodes).

---

[19]See the `supplementary material` ☑ repository for all the query details and their associated data.

[20]A Web-interface to explore the MUSE C-KG is available as `supplementary material` ☑ too.

## F  TAXONOMY CONSTRUCTION OF NLP TASKS IN THE CAPABILITY-KG

In this Appendix, we present how we group various tasks –coming from HuggingFace's Model Cards– into a comprehensive taxonomy of model actions, starting from a seed taxonomy manually curated (see in Figure 5). This taxonomy can be used to structure the PDDL domain files and to have finer domain classification while planning.

Practically, in order to reach a comprehensive and detailed taxonomy, with a useful hierarchy, we started the process by gathering manually various major tasks in NLP. For instance (cf. Figure 5 which represents only an excerpt of the seed taxonomy used), we put `summarisation` and `paraphrasing` within the `text generation` larger category. We then iteratively go through the tasks in the Capability-KG, themselves extracted from HuggingFace Model Cards tasks, and try to integrate them within the taxonomy.

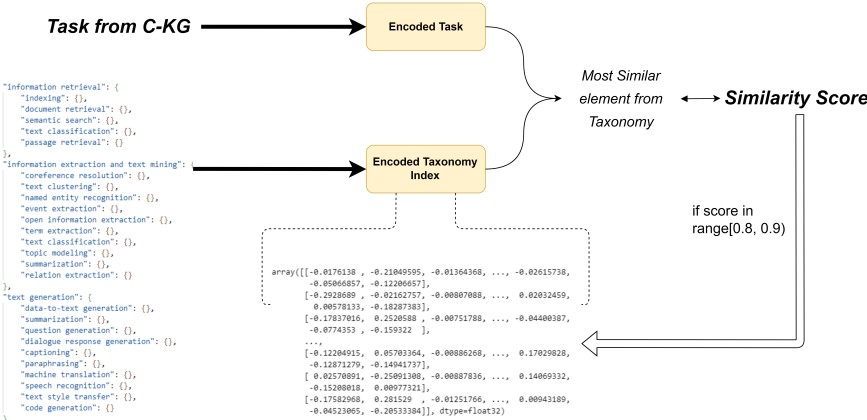

Figure 5: Taxonomy extension protocol: enriching a seed taxonomy with tasks from the C-KG.

The step-by-step process (Figure 5) undertaken can be described as follows:

1. Encode all the elements of NLP taxonomy using Sentence-Transformer
2. Extract all the unique tasks present in the Capability-KG into a list
3. For each task, we search for the most similar element from the taxonomy (and its similarity)
   - if `similarity_score` $\geq$ `0.9`, we consider that the task (or something very close to it) is already present in the taxonomy so we **ignore it**
   - if `0.9 > similarity_score` $\geq$ `0.8`, we consider the task to be relevant and **add it to the taxonomy**
   - if `similarity_score < 0.8`, we ignore the task considering it **irrelevant**

Furthermore, when a new task meets the relevant threshold (`0.9 > similarity_score` $\geq$ `0.8`), it is integrated into the taxonomy under the most similar existing category. This hierarchical structure is traversed to locate the appropriate level, and the new task is added as a sub-category while preserving the nested format. At the end, the taxonomy is flattened to level-3 to maintain a balance between granularity and usability.

Overall, the constructed taxonomy[21] (Figure 6) comprises **420** concepts over 3 hierarchical levels. It goes from high-level conceptual tasks, *e.g.* `syntactic text processing`, to very specific tasks like `morphological tagging`. To the best of our knowledge[22], we are not aware of such a fine-grained taxonomy of NLP tasks focusing on practical applications.

---

[21] Available as a `.json` file in the `supplementary material` ⧉ .
[22] Schopf & Matthes (2024) explored the fields of study in NLP, and even built a classifier; but it does not dive as deep and technical as our taxonomy, but rather uses it to order Academic articles and extract research trends.

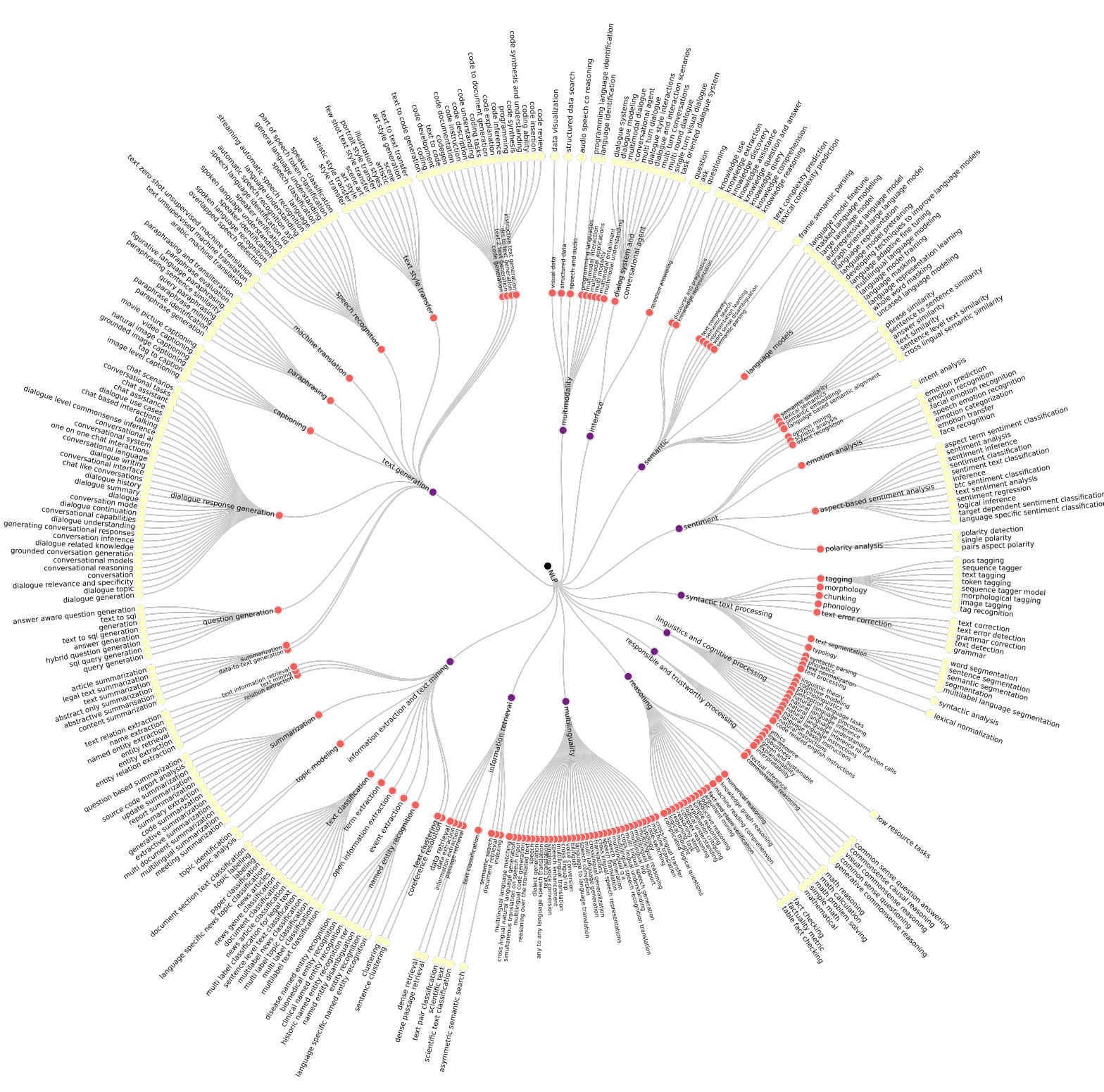

Figure 6: Full taxonomy of the NLP tasks included in the Capability-KG.

