# OpenReview forum: "From an LLM Swarm to a PDDL-empowered Hive: Planning Self-executed Instructions in a Multi-modal Jungle"
_ICLR.cc/2025/Conference — ICLR 2025 Poster_

### Official Review · Reviewer_HTvY · 2024-10-27

**Soundness:** 2
**Presentation:** 2
**Contribution:** 2
**Rating:** 5
**Confidence:** 3

**Summary:**

The paper proposes HIVE, a PDDL-based system to fulfill with user requirements with a large set of foundational models. The system can perform complex actions within given constraints
in an explainable manner.

The paper also proposes a MUSE benchmark to evaluate the multi-modal performance of such agent systems.

**Strengths:**

The papers proposes a method to automatically extract models with capabilities of various modalities from Hugging Face, and explains in details the methodology.

The automatic system also ease the user query but automatically choose the correct model from the pool as the detailed requirement by user in natural language, which is quite meaningful for the rapid-growing multi-modal foundational models in the community.

Experiments show that even the light version can beat the baselines much except for the image-to-audio case. Experiments on the trustworthiness and robustness also show the advantage of HIVE as an explainable system.

**Weaknesses:**

Although an automatic exaction system designed, the number of final candidates for HIVE are unknown. Appendix A only mentions a very small sets of models, most of them fall behind SOTA much (e.g. in image generation, machine translation, text generation). (also see questions)

The paper fails to mention the context why such a system is necessary given that most recent foundational models are for general purpose in its specific modality or multi-modality domain, and user can easily choose a model from public leaderboard or knowledge, if the candidates of HIVE is only a limited set of models which are not necessarily SOTA.

HIVE fails completely (0.0 score) on image-to-audio, 1 of the 6 settings in Table 2, casting doubt on the generalizability of the system.

It's unclear how many modalities and multi-modality settings are covered in main text without looking into appendix. The list shall be small and should be put into main text to give readers a clearer view of the evaluation settings. There also lacks a concrete example of the possible user query HIVE is designed for until section 4.5, making the paper hard to understand.

**Questions:**

+ How many models are selected from Hugging Face as the candidates? Since the community is growing rapidly, will the models from earlier years (with worse performance) be actually selected to perform actions? What's the distribution on model years?
+ It's commonly agreed that recent API-based large models (e.g. GPT-4 series, Gemini) performs better than open-sourced models available on Hugging Face. Have those models been taken into account?
+ Many models are very similar in expertise, since recent community mostly focus on *general*-usage foundational model. What's the importance of such system to choose models from a pool? How many models will be actually selected with common instructions (smallest, best)?
+ Is there any insights why HIVE fails on image-to-audio task? The system should be general enough in modality given the wide availability of multi-modal models, and the 0.0 result can hurt much the overall assessment and actual deployment of HIVE or HIVE-based system.

---

> ### Author Response · Authors · 2024-11-18
> **Answers to comments**
>
> `Although an automatic exaction system designed, the number of final candidates for HIVE are unknown.`
>
> To provide further clarity on the candidate models in the C-KG, here are the details:
>
> - Metadata extracted from HuggingFace: 600,000+ models
> - Models retained in C-KG (based on popularity metrics): 26,806
> - Triples in the C-KG: 125,392
> - Entities in the C-KG: 39,334
>
> Regarding the number of candidates for evaluating HIVE on the MuSE benchmark, we selected 15 models (one per task, as detailed in Appendix A) to manage resource constraints since models needed to be downloaded locally for HIVE and baseline comparisons.
>
> `Appendix A only mentions a very small sets of models, most of them fall behind SOTA much (e.g. in image generation, machine translation, text generation). (also see questions)`
>
> We primarily focus on evaluating the quality of planning, robustness, and trustworthiness of the systems tested. To ensure a fair comparison, we selected smaller models and provided them to all systems—our two Hive configurations, HuggingGPT, and ControlLLM. This approach allows us to assess the core capabilities of each system under consistent conditions.
>
> It's important to note that C-KG includes state-of-the-art (SOTA) models, and if any are not currently present, they can be easily added using our automatic extraction process. However, we do not specifically focus on using SOTA models for execution because our main objectives are to compare the planning capabilities and the ability to select the correct models for tasks based on user preferences. The focus is not on the *quality* of output expected from these execution models. Instead, we concentrate on comparing whether the output *aligns with the user's intent* across all systems. Therefore, the use of SOTA or non-SOTA models for execution does not significantly impact our evaluation.
>
> `The paper fails to mention the context why such a system is necessary given that most recent foundational models are for general purpose in its specific modality or multi-modality domain, and user can easily choose a model from public leaderboard or knowledge, if the candidates of HIVE is only a limited set of models which are not necessarily SOTA.`
>
> Please refer to our overarching note for clarity on our motivation and the response provided to your second question in the following comment, where we discuss performance comparisons with strong OpenAI foundational models.
>
> Selecting an appropriate model is rarely straightforward, particularly for lay-users (such as customers rather than developers), and even for expert practitioners, this task is complex and time-consuming. The decision involves more than just considering benchmark performance; it encompasses a variety of factors like licensing, latency, and resource consumption, which are critical in environments with concurrent parallel services.
>
> Additionally, our complete C-KG includes state-of-the-art models, ensuring that HIVE remains relevant and effective amidst evolving foundational models.
>
> `HIVE fails completely (0.0 score) on image-to-audio, 1 of the 6 settings in Table 2, casting doubt on the generalizability of the system.`
>
> As stated in the conclusion of Section 4.1.1, we acknowledge that image-to-audio conversion is an area we're planning to address in future work. While HIVE currently does not perform well in this specific scenario, it's important to highlight that this is just one out of six settings evaluated.
>
> In the other five cases, HIVE demonstrates strong performance and leads the competition. To further illustrate its generalisability across modalities, we computed the aggregated cross-modalities scores, which are as follows:
>
> | HuggingGPT | ControlLLM | HIVE_light |
> | -- | --- | -- |
> | 0.4320 | 0.2738 | **0.5946** |
>
> These scores indicate that our system generalizes very well across various modalities, despite its current limitations in the niche task of image-to-audio conversion—a task that is also not very common. We are committed to refining HIVE to enhance its capabilities in future updates.
>
> `It's unclear how many modalities and multi-modality settings are covered in main text without looking into appendix. The list shall be small and should be put into main text to give readers a clearer view of the evaluation settings.`
>
> Thank you for this valuable remark. We will ensure that the information on the number of modalities and multi-modality settings is clearly presented in the main text to provide readers with a clearer view of the evaluation settings.
>
> `There also lacks a concrete example of the possible user query HIVE is designed for until section 4.5, making the paper hard to understand.`
>
> Thank you for highlighting this. We will incorporate a concrete example of a possible user query that HIVE is designed for earlier in the paper, ensuring it enhances understanding as readers progress through the sections.

---

> ### Author Response · Authors · 2024-11-18
> **Answers to questions 1, 3 and 4**
>
> `How many models are selected from Hugging Face as the candidates? Since the community is growing rapidly, will the models from earlier years (with worse performance) be actually selected to perform actions? What's the distribution on model years?`
>
> Thank you for your inquiry regarding the selection of models from HuggingFace for our system. To have a better understanding about the number of models in C-KG, please refer to our previous comment. The C-KG currently encompasses a variety of models, as shown in the year-wise distribution table below:
>
> | Year | Number of Models |
> |------|------------------|
> | 2022 | 7,866            |
> | 2023 | 17,574           |
> | 2024 | 1,366            |
>
> We would like to point out that models from earlier years with potentially lower performance may still be considered if final users prioritize other metrics, such as licensing terms or resource consumption, over raw leaderboard performance. The distribution indicates that while most models are from 2023, we have also incorporated some top-performing models from both 2022 and early 2024. As the extraction of the C-KG was conducted in the first quarter, we can see substantial representation of models from this timeframe, maintaining a balanced and comprehensive repository.
>
> `Many models are very similar in expertise, since recent community mostly focus on general-usage foundational model. What's the importance of such system to choose models from a pool? How many models will be actually selected with common instructions (smallest, best)?`
>
> Thank you for your question on the significance of selecting models from a pool. One of our baselines, ControlLLM, maps each task to what it considers the best model. However, this approach can quickly become outdated as new models and criteria emerge. Just keeping these mappings updated isn't enough because "best" varies for each user.
>
> Each user has different needs—some might prioritize licensing, others model size, or leaderboard performance. This is why our approach in HIVE involves selecting from a range of candidate models for each task, allowing flexibility based on user-defined criteria. HuggingGPT also follows a one-to-many selection approach but relies on its core LLM to choose the best candidate.
>
> As for how many models can be selected, we don't just work with common instructions but use structured inputs from users. When users specify different preferences for each task—like licensing or benchmark performance—the number of possible model combinations grows significantly. This ensures users get the most suitable choices tailored to their individual needs. Further explanations are available in Sections 3.2.2 and 4.5.
>
>
> `Is there any insights why HIVE fails on image-to-audio task? The system should be general enough in modality given the wide availability of multi-modal models, and the 0.0 result can hurt much the overall assessment and actual deployment of HIVE or HIVE-based system.`
>
> Regarding why HIVE struggles with image-to-audio tasks, our investigation revealed an issue with the 'parser & rephraser' module. Currently, it doesn't extract instructions properly, affecting the task decomposition step. Despite this, our system still achieves a score of 0.5 on Task Selection (TS) due to its robust design, although unfortunately the Flow of Tasks (FoT) is still misinterpreted, resulting in challenges in this specific case. We are actively working to enhance this module in the future, aiming to improve HIVE's ability to handle diverse modalities even better, further solidifying our generalisability compared to competitors.
>
> For insights related to the image-to-audio task, please refer to the earlier discussion where we explained that HIVE excels in generalisability, outperforming other baselines significantly in overall performance across various modalities.

---

> ### Author Response · Authors · 2024-11-18
> **Answer to question 2**
>
> `It's commonly agreed that recent API-based large models (e.g. GPT-4 series, Gemini) performs better than open-sourced models available on Hugging Face. Have those models been taken into account?`
>
> We may not have fully grasped the question, so we’ll address it from both the possible angles:
>
> ### **Regarding Baselines**:
>    If "into account" refers to considering these large models as baselines (as also suggested by Reviewer XAQH), the recent advancements in models have shown that foundational model capabilities progressed a lot over the past years. This surge may let us believe that at some point in the future only having one-single-large model to-do-everything would be enough. Such a hope, has two issues:
> 1. such a powerful model does not exist yet, and no one knows when it could arise;
> 2. it is barely impossible that such a model could exist and cover all the "niche" use cases and tasks which are currently populating the model landscape (e.g. protein folding or other rare scenarios).
>
> To go further with the first aforementioned point, we ran MuSE over variants of some of the toppest models currently available: o1-preview and GPT-4o both from OpenAI. The results are as follows:
>
> For o1-preview:
> |                        | FS | FoT | O  | fully support |
> | ---------------------- | -- | --- | -- | ------------- |
> | single-domain question | 11 | 11  | 10 | 12            |
> | two-domain question    | 10 | 10  | 11 | 12            |
> | three-domain question  | 5  | 5   | 3  | 5             |
> | total                  | 26 | 26  | 24 | 29            |
>
> Since o1-preview does not support image and audio modality, we only test the cases whose input, output and any expected intermediate result only contains text.
> Overall, `o1-preview` was able to properly answer (O) **24%** of the time and properly justify (TS & FoT) its choices **26%** of the time.
>
> For GPT-4o:
> |                        | FS | FoT | O  | fully support |
> | ---------------------- | -- | --- | -- | ------------- |
> | single-domain question | 16 | 16  | 15 | 20            |
> | two-domain question    | 24 | 24  | 23 | 24            |
> | three-domain question  | 4  | 4   | 5  | 7             |
> | total                  | 44 | 44  | 43 | 51            |
>
> Once again, since `GPT-4o` only support text and image input and textual output, we filtered out all the entries involving unsupported modalities.
> Overall, `GPT-4o` was able to properly answer (O) **43%** of the time and properly justify (TS & FoT) its choices **44%** of the time.
>
> Unsurprisingly, even if both are able to plan and to perform over MuSE for some of its queries, both the best OpenAI foundational models are not yet able to deal with the richness of real-world multi-modal scenarios as depicted in MuSE.
>
>
> ###  **Incorporation in Execution Pools**:
>    If "into account" refers to these models being part of the execution pool, our focus has not been on execution model performance. Instead, it's been on evaluating how well the tested solutions can plan complex, multi-task user queries. Therefore, while SOTA execution models are significant (and included in our C-KG as stated in the previous comment), our primary evaluation criterion is on planning capability.

---

> ### Comment · Reviewer_HTvY · 2024-11-21
> **Feedback to Author Rebuttals**
>
> Thanks for the clarification about the number and year distribution of the models involved in HIVE. I think I misunderstood the scale of the number of models when I was writing the initial review.
>
> Thanks for the analysis of why HIVE fails on image-to-audio task. I think the explanation makes sense. I suggest author to include this in paper/appendix to help avoid concern of generalizability.
>
> Thanks for the clarification and evidence that frameworks like HIVE are necessary even with recent very strong foundational model.
>
> However, I still have a concern about the 20k models in HIVE framework. Most actual usage of HIVE will be performance-based or efficiency-based. Also the author mentions some other factors users may take consideration, I still doubt the actual effective portion of the collected models, which may be chosen in a regular use case.
>
> Since much of my concerns has been addressed, I have increased my score from 3 to 5.

---

> > ### Author Response · Authors · 2024-11-27
> > **Further Clarifications**
> >
> > Thank you for updating your score—we greatly appreciate your thorough review and thoughtful feedback on our rebuttal. We respectfully argue that monitoring capabilities across a plethora of models, potentially extending beyond 20,000, is crucial in today's evolving landscape.
> >
> > Recent research indicates that valuable complementary knowledge can be derived from multiple models, even for tasks covered by benchmarks traditionally considered **solved** (see [Roth _et al._, ICLR, 2024](https://arxiv.org/pdf/2310.17653)). This is particularly relevant because even top-performing models may fail on specific data instances where others succeed, demonstrating that no single model performs optimally across all scenarios. Moreover, updated versions of models do not always guarantee superiority on every instance. Prior model versions can still outperform their successors in specific cases, as highlighted by recent findings (see [Apple ML Research, EMLNP, 2024](https://arxiv.org/pdf/2407.09435)).
> >
> > In this context, we argue that our Capability-KG can be detrimental in monitoring model performances at such data-instance-level of granularity, which could easily scale up to several thousands of models if you consider different, versions/combinations, and user-specific constraints such as licensing and inference costs.
> >
> > We believe this granular approach ensures that the system remains adaptable, comprehensive, and efficient in diverse, real-world applications.

---

### Official Review · Reviewer_MPzp · 2024-11-03

**Soundness:** 3
**Presentation:** 3
**Contribution:** 3
**Rating:** 6
**Confidence:** 3

**Summary:**

This paper proposes a new system, HIVE, that coordinates instructions and task executions across models. HIVE also explains steps being taken. HIVE incorporates a capability knowledge graph to organize and retrieve model capabilities and it uses PDDL, a formal logic approach to assist with planning. To thoroughly evaluate, they also developed a new multi-modal benchmark, MuSE, consisting of complex, multi-modal queries. They demonstrate that HIVE outperforms competitors such as HuggingGPT and ControlLLM on MuSE.

**Strengths:**

Originality
-They claim to be the first multiple model approach to leverage PDDL.
-Their approach consists of intuitive steps, achieved using new ideas such as PDDL for planning and C-KG for model selection.
-The multi-modal benchmark data set appears to be a new contribution in the multiple model task domain. More clarity here would be great, see my comments below.

Quality
-The idea of using model cards systematically to leverage new models is very practical and useful. I can see this being helpful for doing this at scale (say you have 1000+ models from which to choose).
-HIVE outperforms the competing models on their more challenging dataset (MuSE)

Clarity
-Overall, the steps in their system: parse user prompt → plan → select models → execute is clear and intuitive.

Significance
-They solve the problem of model selection based on many constraints: hardware (model size), licensing, and model performance. This is going to be increasingly important for multi-agent/multiple model systems.

**Weaknesses:**

-PDDL was not defined until later in the paper - define it in the abstract and/or introduction.
-Not clear if MuSE will be openly available
-The interaction between PDDL and LLM (ChatGPT) in task decomposition could be made clearer - what exactly is the LLM doing and how is it leveraging PDDL? I think since PDDL could be new for readers, being clear here is very important.

**Questions:**

-I suggest making the abstract more crisp and making the distinction of this work very clear right away.

-Figure 2 needs to be cleaned up - time is in seconds? The numbers look crowded in the image.

-How does this dataset compare with the function-calling datasets? Is the key difference that these queries require model calls as solutions (rather than API/function calls)?

-Will MuSE be open source?

-Are there multi-modal query-trajectory datasets in the computer vision community? If so, how do those compare with your multi-modal dataset?

---

> ### Author Response · Authors · 2024-11-15
> **Answers to comments**
>
> `- PDDL was not defined until later in the paper`
> `- define it in the abstract and/or introduction.`
>
> Thank you for your suggestion. We'll ensure that PDDL is defined in the abstract and introduction to enhance clarity and provide context right from the beginning.
>
> `- Not clear if MuSE will be openly available`
>
> Currently, MuSE is included in the Supplementary Material due to the double-blind review process. However, it will be openly available upon acceptance.
>
> `- The interaction between PDDL and LLM (ChatGPT) in task decomposition could be made clearer`
> `- what exactly is the LLM doing and how is it leveraging PDDL? I think since PDDL could be new for readers, being clear here is very important.`
>
> Thank you for highlighting this. We recognize that PDDL might be unfamiliar to the NLP community. We'll enhance the explanation in Section 3.2.1 and have also revised the overall architecture diagram for Hive. (We will soon upload a revision of the article containing the revised Figure 1.)

---

> ### Author Response · Authors · 2024-11-15
> **Answers to questions**
>
> `-I suggest making the abstract more crisp and making the distinction of this work very clear right away.`
>
> Sure,  we'll revise the abstract to ensure it is more concise and clearly highlights the distinct contributions of our work.
>
> `-Figure 2 needs to be cleaned up - time is in seconds? The numbers look crowded in the image.`
>
> Certainly. We will refine the TikZ/PGFplot to improve clarity, and we will add "seconds" to the x-axis as well as the caption
>
> `-How does this dataset compare with the function-calling datasets? Is the key difference that these queries require model calls as solutions (rather than API/function calls)?`
>
> Thank you for your question regarding the comparison of MuSE with the existing function-calling datasets.
> The key distinction lies in the nature of the actions required to generate the expected output. Indeed, while function-calling datasets [1,2] primarily focus on generating a sequence of API signatures where arguments are filled with values extracted from either the user query or additional API calls, our dataset, MuSE, specifically targets actions that involve calls to LLMs. From an implementation point of view, this is achieved by executing a Python script which loads the relevant models, performs preprocessing steps (text cleaning, tokenization, etc), conducts inference, and finally outputs the result. This approach allows to handle much more complex user queries, as opposed to strictly relying on API calls. We hope this clarification helps to highlight the differences and uniqueness our dataset compared to function-calling ones.
>
> _[1] Qin, Y., Liang, S., Ye, Y., Zhu, K., Yan, L., Lu, Y., Lin, Y., Cong, X., Tang, X., Qian, B. and Zhao, S., 2023. Toolllm: Facilitating large language models to master 16000+ real-world apis. arXiv preprint arXiv:2307.16789._
> _[2] Patil, S.G., Zhang, T., Wang, X. and Gonzalez, J.E., 2023. Gorilla: Large language model connected with massive apis. arXiv preprint arXiv:2305.15334._
>
> `-Will MuSE be open source?`
>
> Yes.
>
> `-Are there multi-modal query-trajectory datasets in the computer vision community? If so, how do those compare with your multi-modal dataset?`
>
> Our primary aim with MuSE is to demonstrate the planning capabilities of the Hive system across various task combinations and modalities, rather than focusing solely on the vision domain.
> MuSE fills this gap by enabling the evaluation of complex, integrated user queries that encompass multiple tasks, which is essential for real-world applications. For instance, consider a user query requiring the processing of an image with VQA, generating a summary and identifying named entities through NER. Standard benchmarks can't naturally merge to evaluate such tasks without significant manual intervention to customize user instructions and step sequences. MuSE inherently supports this workflow, allowing for a more comprehensive evaluation of a model's adaptability and effectiveness when handling compound tasks, thus providing a clearer reflection of actual user requirements and multi-disciplinary demands. If you have specific datasets in mind from the computer vision community, please share them with us, and we'd be happy to review and outline the main differences.

---

### Official Review · Reviewer_XAQH · 2024-11-06

**Soundness:** 2
**Presentation:** 2
**Contribution:** 2
**Rating:** 5
**Confidence:** 3

**Summary:**

The paper presents Hive, an agent-based system that selects and coordinates models to execute complex tasks based on user instructions. Hive creates explainable plans using atomic actions across various models, guided by formal logic and PDDL operations, ensuring transparency and adherence to user constraints. The authors also introduce the MuSE benchmark to evaluate multimodal capabilities in agent systems. Results show that Hive outperforms other systems in task selection and transparency.

**Strengths:**

The KG construction method is thorough, utilizing model metadata from multiple sources to enhance graph quality. The entire planning pipeline demonstrates a clear improvement over prior work, such as HuggingGPT. Additionally, the writing is well-structured and easy to follow, which supports reader comprehension.

**Weaknesses:**

This paper has several notable weaknesses:

First, it feels like the approach relies heavily on leveraging a powerful LLM to decompose user tasks based on key attributes, selecting open-source models from Huggingface, and executing plans sequentially. In essence, the paper presents a tool-usage framework where the tools are Huggingface models. While there are some unique designs for processing and indexing these models, the contribution appears marginal. The problem-solving paradigm closely resembles prior works, such as HuggingGPT. Additionally, the paper lacks baseline comparisons, particularly in evaluating why their proposed KG construction method is optimal or how it could be further applied in other scenarios.

The experimental design raises concerns as well. Although the authors introduce a new evaluation benchmark, it is based on an in-house dataset. It’s unsurprising that the method performs well on this dataset, but it lacks validation on standard benchmarks, such as classical VQA datasets for multimodal evaluation.

On a broader level, the necessity of maintaining a Huggingface model graph is questionable. As more cheap and advanced models (e.g., GPT-5, LLaMA-4, lower-cost GPT-4o) emerge with capabilities for comprehensive task handling, the reliance on a graph of smaller, domain-specific models may become obsolete. While a graph of non-LLM-based APIs remains valuable (e.g., calculators outperforms LLMs in calculation), most small LLM functionalities in the current graph could eventually be covered by future, more capable LLMs. This brings into question the core motivation behind the paper’s studied topic.

**Questions:**

Could you please test the pipeline or more baselines on more widely used benchmarks in multimodal domains?

Would you provide more examples about how the KG looks like and an entire planning example from scratch?

---

> ### Author Response · Authors · 2024-11-15
> **Answers to comments [1/2]**
>
> ` First, it feels like the approach relies heavily on leveraging a powerful LLM to decompose user tasks based on key attributes, selecting open-source models from Huggingface, and executing plans sequentially. `
>
> Not really, _Hive light_ uses 7b-8bit-quantized models and _Hive_ uses GPT-3.5, both already beat SotA in our experiments.
>
> `While there are some unique designs for processing and indexing these models, the contribution appears marginal. The problem-solving paradigm closely resembles prior works, such as HuggingGPT.`
>
> Indeed, the **motivations** are similar _i.e._ leveraging a set of models to answer user queries. However, there are major differences between the considered baselines (namely HuggingGPT and ControlLLM):
> 1. a structured language (PDDL) powered Hive's planning and brings logic into the reasoning while other rely solely on inference/prompting via LMs (e.g. CoT, GoT, …);
> 2. a deterministic KG-based model selection which also takes into account the specificities defined by each single user.
>
> `The experimental design raises concerns as well. Although the authors introduce a new evaluation benchmark, it is based on an in-house dataset. It’s unsurprising that the method performs well on this dataset, but it lacks validation on standard benchmarks, such as classical VQA datasets for multimodal evaluation.`
>
> To better answer the above comment, we split it in two aspects as follows.
>
> **1] Why did we have to build a new dataset from scratch?**
> First of all, we could not rely (or build upon) the HuggingGPT's or ControlLLM's validation routines presented in their respective papers, as they did NOT share their benchmarks.
> This fact led us to consider searching for existing state-of-the-art benchmarks (or at least datasets) covering or needs: a dataset providing natural language instructions associated with various modalities as inputs and expecting, as outputs, various ones too. The main idea behind building a benchmark was that we wanted to have cross-modalities in multi-tasked queries.
> As a consequence, in order to validate our approach we had no choice but constructing complex evaluation scenarios.
> In addition, MuSE would be released publicly (waiting now for double-blind policy but in the Supplementary Material for the time being), letting therefore the community rely on it for future efforts.
>
> **2] Why is MuSE better suited than a conjunction of "standard benchmarks"?**
> MuSE is better suited than standard benchmarks because it focuses on multi-tasked query evaluation, unlike traditional benchmarks which assess single-task performance (e.g., VQA, Text Generation, ASR). Standard benchmarks excel in measuring individual task efficacy but fall short in assessing a model's ability to perform tasks that require combining different capabilities. Moreover, instead of only assembling single-task queries (e.g. summarise A and classify B), as one may achieve by combining multiple standard benchmarks, MuSE also expect using the intermediate result of one task as an input for another (e.g. summarise A and classify the summary).
>
> MuSE fills this gap by enabling the evaluation of complex, integrated user queries that encompass multiple tasks, which is essential for real-world applications. For instance, consider a user query requiring the processing of an image with VQA, generating a summary and identifying named entities through NER. Standard benchmarks can't naturally merge to evaluate such tasks without significant manual intervention to customize user instructions and step sequences. MuSE inherently supports this workflow, allowing for a more comprehensive evaluation of a model's adaptability and effectiveness when handling compound tasks, thus providing a clearer reflection of actual user requirements and multi-disciplinary demands.

---

> ### Author Response · Authors · 2024-11-15
> **Answers to comment [2/2]**
>
> `On a broader level, the necessity of maintaining a Huggingface model graph is questionable. As more cheap and advanced models (e.g., GPT-5, LLaMA-4, lower-cost GPT-4o) emerge with capabilities for comprehensive task handling, the reliance on a graph of smaller, domain-specific models may become obsolete. While a graph of non-LLM-based APIs remains valuable (e.g., calculators outperforms LLMs in calculation), most small LLM functionalities in the current graph could eventually be covered by future, more capable LLMs. This brings into question the core motivation behind the paper’s studied topic.`
>
> The recent advancements in models have shown that foundational model capabilities progressed a lot over the past years. This surge may let us believe that at some point in the future only having one-single-large model to-do-everything would be enough. Such a hope, has two issues:
> 1. such a powerful model does not exist yet, and no one knows when it could arise;
> 2. it is barely impossible that such a model could exist and cover all the "niche" use cases and tasks which are currently populating the model landscape (e.g. protein folding or other rare scenarios).
>
> To go further with the first aforementioned point, we ran MuSE over variants of some of the toppest models currently available: **o1-preview** and **GPT-4o** both from OpenAI. The results are as follows:
>
> For o1-preview:
> |                        | FS | FoT | O  | fully support |
> | ---------------------- | -- | --- | -- | ------------- |
> | single-domain question | 11 | 11  | 10 | 12            |
> | two-domain question    | 10 | 10  | 11 | 12            |
> | three-domain question  | 5  | 5   | 3  | 5             |
> | total                  | 26 | 26  | 24 | 29            |
>
> Since o1-preview does not support image and audio modality, we only test the cases whose input, output and any expected intermediate result only contains text.
> Overall, o1-preview was able to properly answer (O) **24%** of the time and properly justify (TS & FoT) its choices **26%** of the time.
>
> For GPT-4o:
> |                        | FS | FoT | O  | fully support |
> | ---------------------- | -- | --- | -- | ------------- |
> | single-domain question | 16 | 16  | 15 | 20            |
> | two-domain question    | 24 | 24  | 23 | 24            |
> | three-domain question  | 4  | 4   | 5  | 7             |
> | total                  | 44 | 44  | 43 | 51            |
>
> Once again, since GPT-4o only support text and image input and textual output, we filtered out all the entries involving unsupported modalities.
> Overall, GPT-4o was able to properly answer (O) **43%** of the time and properly justify (TS & FoT) its choices **44%** of the time.
>
> Unsurprisingly, even if both are able to plan and to perform over MuSE for some of its queries, both the best OpenAI foundational models are not yet able to deal with the richness of real-world multi-modal scenarios as depicted in MuSE.
>
> More generally, and in order to tackle the point _2_ listed above about the "niche" tasks, we would like to remind that the essence of the Capability-KG lies in the fact that its richness allows to retrieve candidate models for a large set of tasks (more than 100 at the moment) and does not limit users to envision instructions revolving around a handful of popular tasks (such a point as been highlighted by `Reviewer MPzp`'s comment "This is going to be increasingly important for multi-agent/multiple model systems").
>
> Finally, we would like to mention a direction we are currently exploring. So far, the presented C-KG gathers information related to models, but structurally, nothing prevents us from collecting and structuring information related to other _objects_. For instance, the graph nature of the C-KG led us starting the exploration of having sub-nodes for adapters attached to main-principal-models; these adapter-nodes still carry the same type of information as the main nodes currently listed in the C-KG, gathering performance information for instance. In the future, we think this addition would also allow building complex instruction pipeline which would rely mainly on a single backbone model and this could be a solution for some specific actors having access to only one model.
>
> More generally, we would wrap up this comment-answer, emphasising on the fact that even if the big models available at the moment aren't yet capable like a group of selected specialised models, the overall technical architecture of Hive is still appropriate to tackle scenarios where only one model and associated variants (through adapters for instance) is available.

---

> ### Author Response · Authors · 2024-11-15
> **Answers to questions**
>
> `Could you please test the pipeline or more baselines on more widely used benchmarks in multimodal domains?`
>
> As there are no standard benchmark covering our use-case (as mentioned in the in-line comment answer), and as there are no baseline that tackle the same challenge as us apart from HuggingGPT and ControlLLM, we evaluated MuSE on the latest OpenAI models (for which you can see the results integrated above) to support the intuition that currently most-advanced models are not yet able to comprehensively handle complex instructions/tasks.
>
> `Would you provide more examples about how the KG looks like and an entire planning example from scratch?`
>
> Please see **Screencast.mp4** (for a complete run through in real time) and the browser interface **C-KG_excerpt/capability_kg.html** (for the C-KG excerpt used in MuSE) in the supplementary material .zip archive.

---

### Author Response · Authors · 2024-11-15
**Overall Note**

Dear Reviewers,

**1)** We would like to express our gratitude for your time and careful consideration of our paper.

**2)** We have noted the common feedback regarding the clarity of the article's main motivation and claims. To facilitate further discussion, here is a clarification:

> The primary aim of HIVE is to provide a unified and explainable solution for responding to natural language (NL) instructions that may involve multi-modal inputs and expect multi-modal outputs. The system respects user conditions across a variety of metrics, not solely performance-based ones. This necessitates distributing sub-instructions to potentially different and specialized models. To achieve this, we structured the landscape of available models to select the most suitable one for each task, utilizing our C-KG. Addressing the explainability challenge, we chose to shift the planning actions from the model space—where competitors focus—to the formalism of PDDL, a planning language widely used in robotics. This transition enables us to clearly understand the steps HIVE takes to formulate an answer.

> Additionally, we introduced a new benchmark, MuSE, because existing baselines do not provide a shared benchmark. MuSE enables the evaluation of complex, integrated user queries encompassing multiple tasks, crucial for real-world applications. Recognizing the lack of standard public benchmarks for such tasks, we constructed MuSE around some of the most popular tasks (15 in total). While not holistic, it effectively assesses our proposed framework, and we welcome future contributions to expand and enhance the benchmark.

As recommended by `Reviewer MPzp`, we intend to revise the Introduction and abstract to emphasize our motivation more effectively.

**3)** To provide comprehensive responses, we have organized our feedback into dedicated posts addressing each reviewer’s comments and questions.

Thank you again for your insightful feedback.

---

### Author Response · Authors · 2024-11-27
**Revised Article**

Dear ACs, SACs, PCs, and Reviewers,

We appreciate the opportunity to address the reviewers' feedback and hope our responses have clarified any concerns raised. Below, we highlight the key strengths of our work as acknowledged by the reviewers and outline the improvements made in the revised version of our article.

---

### **Key Strengths of Our Method:**

1. **Innovative Knowledge Graph Construction:**
   Our approach integrates model metadata from diverse sources, significantly enhancing the quality and depth of the resulting knowledge graph. The proposed planning pipeline demonstrates substantial improvements compared to previous works, such as HuggingGPT.

2. **Novel task planning:**
   The systematic application of PDDL for planning with LLMs, combined with the C-KG framework for model selection, represents a novel contribution in the multi-model task domain. Reviewers particularly highlighted the originality of using PDDL in this context, along with the introduction of a comprehensive multi-modal benchmark dataset.

3. **Clear and Intuitive System Structure:**
   The system’s structured approach—encompassing user prompt parsing, planning, model selection, and task execution—efficiently addresses complex model selection challenges. By incorporating constraints like hardware availability, licensing, and performance, our method advances the state of multi-agent systems. Additionally, leveraging model cards for integrating new models ensures scalability, which is crucial when navigating extensive model libraries.

4. **Robust Experimental Results:**
   Experiments conducted using MuSE demonstrate that even a lightweight version of our system outperforms existing state-of-the-art solutions in most scenarios. The HIVE framework, in particular, has shown notable trustworthiness and robustness.

---

### **Revisions Made:**

In response to the feedback received, we have revised our manuscript and uploaded the updated version. The key modifications include:

- Rephrased the **Abstract** and **Introduction** for clarity and conciseness.
- Added a bit of more background related to **PDDL** in the introduction.
- Enhanced **Footnote 10** to include the multi-modality settings of MuSE.
- Restructured **Appendix B** for better readability.
- Added section references in **Appendix C**.
- Included **C-KG metadata** in Appendix D.
- Redesigned **Figure 1** and improved **Figure 2** for better visual representation.

---

We sincerely appreciate your time and effort in reviewing our work. Your valuable feedback has been instrumental in strengthening our contribution.

Thank you and best regards,
—
The Authors

---

### Meta-Review · Area_Chair_duSp · 2024-12-19

**Metareview:**

The paper presents a method to use many small (possibly domain specific) models to execute low level actions in an environment. While this concept of high level planning to low level actions with LLMs is not new by itself, the paper does appear to situate itself well with the existing literature and proposes a seemingly novel method with reasonable results.

Many of the reviewers' concerns that are not directly addressed by the authors are rather ideological in nature ("why bother with this line of work using many domain specific models if hypothetical models like GPT-5 or Llama 4 can just do it all one day" is one example that stood out to me) and are either invalid or not relevant to the topic of this paper's contributions to the current state of AI.

**Additional Comments On Reviewer Discussion:**

I am discounting the review by XAQH and MPzp  as there was no engagement during the rebuttal. The rest of the reviewers' concerns seem to have been sufficiently addressed and many of the remaining differences are more ideological than anything as mentioned in the main metareview.

---

### Decision · Program_Chairs · 2025-01-22

Accept (Poster)